# Curbing household food waste and associated climate change impacts in an ageing society

Yosuke Shigetomi [1,2] ✉, Asuka Ishigami[3], Yin Long[4] & Andrew Chapman [5]

We explored the intricate quantitative structure of household food waste and their corresponding life cycle greenhouse gas emissions from raw materials to retail utilizing a combination of household- and food-related economic statistics and life cycle assessment in Japan. Given Japan's status as a nation heavily impacted by an aging population, this study estimates these indicators for the six age brackets of Japanese households, showing that per capita food waste increases as the age of the household head increases (from 16.6 for the 20's and younger group to 46.0 kg/year for 70's and older in 2015) primarily attributed to the propensity of older households purchase of more fruits and vegetables. Further, the largest life cycle greenhouse gases related to food waste was 90.1 kg-$CO_2$eq/year for those in their 60's while the smallest was 39.2 kg-$CO_2$eq/year for 20's and younger. Furthermore, food waste and associated emissions are expected to decline due to future demographic changes imparted by an aging, shrinking population after 2020 until 2040. Specific measures focused on demographic shifts are crucial for Japan and other countries with similar dietary patterns and demographics to achieve related sustainable development goals through suppressing food waste and associated emissions under new dietary regimes.

The process of food consumption bears profound significance not merely for humankind but also for planetary well-being[1]. However, our everyday food needs necessitate substantial energy and natural resources throughout the supply chain, in turn causing significant reverberations on global environmental sustainability from a life-cycle perspective[2–5]. Using life cycle assessment (LCA), it was found that the direct and indirect greenhouse gas (GHG) emissions associated with the international food chain accounted for more than one-third of total global GHG emissions, namely 17 giga tons ($10^9$ ton; Gt) of carbon dioxide equivalent ($CO_2$eq) and that these emissions increased by ~2.2 Gt-$CO_2$eq from 1990 to 2018 considering land-use and land-use changes (LULUC) resulting from food production[6]. Further, global water consumption, deforestation, and biodiversity losses led by

LULUC in response to food consumption in developed nations have also been occurring in developing nations and are of high concern[7–10]. Given this context, the minimization of unnecessary food production globally emerges as a pivotal strategy towards improving both food security and environmental sustainability[4,11–14].

Nevertheless, approximately one-third of the food produced globally is not consumed; i.e., lost, or wasted, accounting for about 1.3 Gt[15]. This accounts for 17% of total food production[16]. Wastage also contributes to approximately half of global food supply chain GHG emissions (i.e., 9.3 Gt-$CO_2$eq)[17] and are worth ~1 billion US dollars in environmental and social costs[18]. Hence, the suppression of food loss and waste along the international supply chain is regarded as an issue of environmental, social, and economic sustainability, targeted under

[1]Faculty of Science and Engineering, Ritsumeikan University, 1-1-1 Nojihigashi, Kusatsu, Shiga 525–8577, Japan. [2]Research Institute for Humanity and Nature, 457-4 Motoyama, Kamigamo, Kita-ku, Kyoto 603-8047, Japan. [3]Faculty of Environmental Science, Nagasaki University, 1-14 Bunkyo-machi, Nagasaki 852-8521, Japan. [4]Graduate School of Engineering, The University of Tokyo, 7-3-1 Hongo, Bunkyo-ku, Tokyo 113-8654, Japan. [5]International Institute for Carbon Neutral Energy Research, Kyushu University, 744 Motooka, Nishi-Ku, Fukuoka 819-0395, Japan. ✉e-mail: y-shig@fc.ritsumei.ac.jp

Sustainable Development Goal (SDG) 12[19]. Within each supply chain of food, household consumers are a key part of the issue, particularly in economically developed nations[20–23]. Thus, there is a need for consumers to consider additional efforts to reduce their food waste by avoiding specific behaviors at home such as preparing too much food or leaving food on their plates. At the same time, policies to guide these behaviors should be developed to facilitate food labeling (e.g., "best before" vs "use by"), guidelines, and food banks[24,25].

Japan, despite being a highly economically developed nation, relies heavily on food imports from foreign countries[26]. Due to this high dependency on imports, 34% of life cycle GHG emissions due to Japanese food consumption were induced overseas[27]. Nevertheless, the latest ministry report estimated that 22% of total food loss and waste generated in the domestic supply chain, some 5.23 megatons (Mt), were likely still edible and 47% (2.47 Mt of food waste) was generated from households in 2021[28]. In response, the Government of Japan has been committed to suppressing household food waste at 50% of year 2000 levels by fiscal year (FY) 2030 in the Fourth Fundamental Plan for Establishing a Sound Material-Cycle Society[29]. A recent domestic study depicted the structure of overall food waste and related GHG emissions in Japan[30]. However, how much, what foods, and by whom, each Japanese household's food waste and related GHG emissions are engendered has not been clarified.

Our research aims to identify the structure of household food waste (FW) and its associated life cycle GHG (FWGHG) in Japan. Critically, we consider FW and FWGHG in light of differing consumption patterns segmented by the age group of the household head. Through this focused approach, this study endeavors to illuminate the path toward food-related decarbonization actions within the context of an aging society, taking Japan as a case study due to its worldwide leading ratio of elderly population (those aged 65 and above) which accounts for 29.8% of the total population, the second highest ratio among all nations in 2021[31]. A demographic transition towards an aging society tends to influence food-related GHG emissions[32,33]. In addition, Japan, compared to similarly developed nations such as the US, the Netherlands, Australia and New Zealand has a relatively high level of food waster per capita[16]. Hence, we elucidate the relationship between household FW, FWGHG, and dietary trends cognizant of age. In this study, household FW is defined as the amount of edible parts of food that are directly discarded by households, i.e., not including FW from foods consumed in restaurants and as takeout meals, and in school lunches etc.[34]. FWGHG covers the emissions created from cradle to retail (not including emissions related to cooking). Further, we discuss policy strategies for the reduction targets of FW committed to by the government, cognizant of the diversity of food consumption by age bracket and future demographic trends including an aging, shrinking population. The findings are relevant to many other nations experiencing similar demographic changes and increased consumption of hyper-convenient and ultra-processed foods[35], and a decreased adoption of traditional diets[36]. Through our findings and discussion, we document insights toward a reduction in FW and FWGHG and achievement of related SDGs cognizant not only of food wastage, but also of impact exacerbation due to an aging society.

## Result

### Total household food waste and household food waste per capita by age bracket for Japan

The composition of total household FW for Japan in 2015 was calculated as detailed in Fig. 1. Here, 197 food consumption items are aggregated into 11 categories according to major divisions established from Japanese consumer expenditure surveys (Family Income Expenditure Survey (FIES)[37] and National Survey of Family Income and Expenditure (NSFIE)[38], see Method section; "grains," "fishery and seafood," "meats," "dairy products," "vegetables," "fruits," "oils and seasoning," "confectionary," "ready meals," "soft drink," and "alcohol").

The total amount of household FW was 2.89 Mt/yr. Overall, the "vegetables" category has the largest contribution to total FW, accounting for 43% of the total (i.e., 1.23 Mt/yr). In this category, *cabbage* (hereafter, the name of the detailed commodity is written in Italics) was the largest contributor to FW, accounting for 0.12 Mt/yr. The next most impactful vegetables were *other leafy greens* (e.g., Chinese cabbage), *onion*, and *tomatoes*, contributing 0.11, 0.10, and 0.093 Mt/yr, respectively. The second largest category of FW was "fruits," and its main contributors were *banana*, *apple*, and *tangerine*, contributing 0.097, 0.083, and 0.060 Mt/yr, respectively. Surprisingly, the FW imparted from these two categories was 57% of all FW. For other categories, *rice* (0.070 Mt/yr), *other cooked meal* (0.058 Mt/yr), *milk* (0.086 Mt/yr), *lunchbox* (0.046 Mt/yr), and *egg* (0.042 Mt/yr) were estimated to generate relatively large amounts of FW.

The structures of mean per-capita household FW by age bracket are depicted in Fig. 1b. Overall, per-capita FW was likely to increase as people aged. The highest per-capita FW was generated by households in their 70 s and older, some 46.0 kg/cap·yr, slightly higher than those in their 60 s (44.4 kg/cap·yr). These per-capita FW levels were more than twice those of households in their 20 s and younger, generating 16.6 kg/cap·yr. These discrepancies in FW between age brackets may arise due to different dietary preferences (Fig. 1b); fresh meals such as "vegetables," "fruits," and "fishery and seafood," were purchased by older households in particular. Figure 1c depicts how much FW was generated by excessive food preparation (i.e., too much cutting), disposal, and leftovers, showing traits among age brackets. As seen in the figure, excessive preparation was the main reason for the total FW for all of the age brackets except those in their 20 s, followed by leftovers and disposal. Younger households were less likely to generate FW due to excessive preparation, particularly those in their 20 s and younger who generated their FW predominantly from leftovers (43% of their FW). On the other hand, the proportion of FW generated by excessive preparation markedly increased as the household head got older (from 34 – 50% of their FW). Direct disposal were almost stable at 20% among young and elderly households.

### Total GHG emissions related to household food waste and those emissions by age bracket for Japan

The total FWGHG was quantified to be 6.06 Mt-$CO_2$eq/yr (Fig. 2a). The essential drivers of FWGHG were different to those for the composition of FW. The largest category contributing to FWGHG was "vegetables" followed closely by "ready meals", "fishery and seafoods", and "meats". GHG impacts were estimated to be 1.28, 1.13, 0.70, 0.67 Mt-$CO_2$eq/yr, respectively. The FWGHG of "fruits" and "grains" were similar, around 0.45 Mt-$CO_2$eq/yr. Specifically, *other cooked meal* (0.38 Mt-$CO_2$eq/yr), *beef* (0.22 Mt-$CO_2$eq/yr), and *other bread* (0.20 Mt-$CO_2$eq/yr) were the largest three commodities underpinning FWGHG, followed by *pork* (0.17 Mt-$CO_2$eq/yr), *other mushrooms* (0.16 Mt-$CO_2$eq/yr), and *milk* (0.15 Mt-$CO_2$eq/yr). Within "vegetables" and "fruits", *tofu* (0.10 Mt-$CO_2$eq/yr), *strawberry* (0.10 Mt-$CO_2$eq/yr), *cucumber* (0.078 Mt-$CO_2$eq/yr), and *banana* (0.076 Mt-$CO_2$eq/yr) are notable for their contributions to FWGHG. These food commodities are also likely to spoil easily.

When considering the structures of the mean per-capita FWGHG (Fig. 2b, c), households in their 60 s generated slightly higher emissions than those in their 70 s. Both accounted for around 90 kg-$CO_2$eq/cap·yr. The per-capita FWGHG for those in their 20 s and younger was 39.2 kg-$CO_2$eq/cap·yr, the lowest among age brackets. Compared to the case of FW, both contributions of leftovers and disposal to FWGHG were larger than those arising from excessive preparation. This finding implies that strategies for mitigating FW and FWGHG would differ for people at different life stages.

Figure 3 illustrates the impacts toward both FW and FWGHG for 28 food groups; the size of the box implies the effectiveness of reducing FW on avoided GHG emissions created through the production

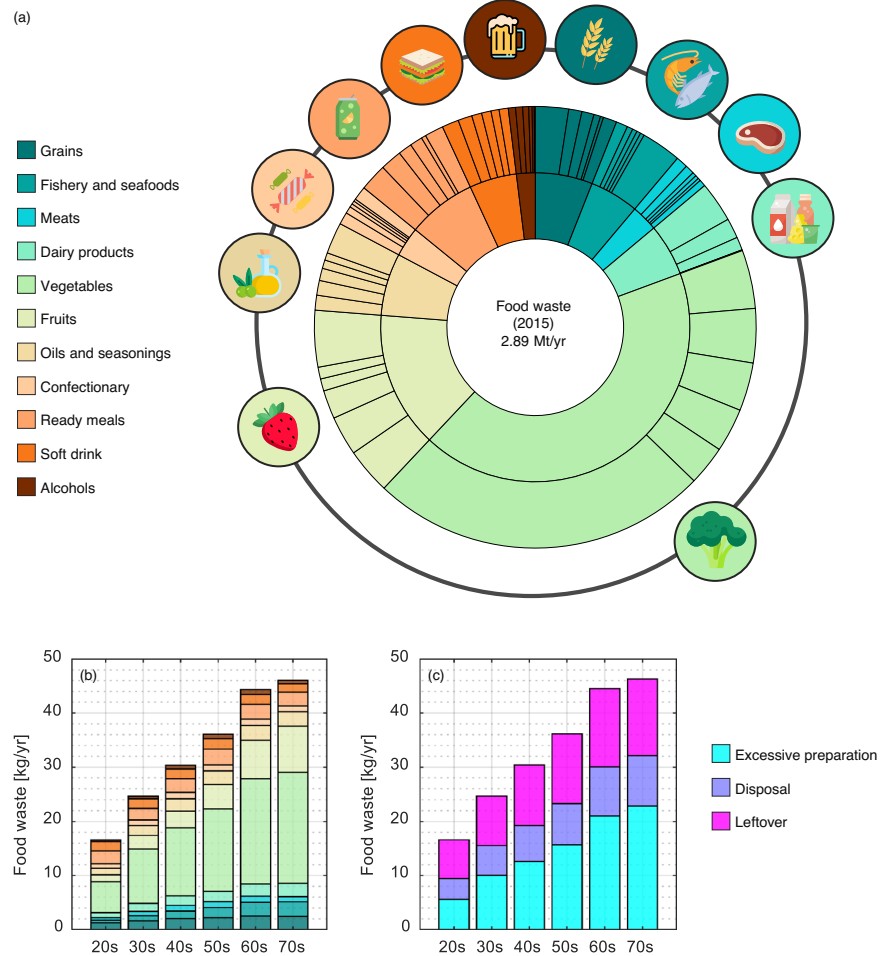

**Fig. 1 | Compositions of total household food waste (FW) for Japan in 2015.** The total FW is shown by 11 aggregated categories from 197 food commodities in the inner circle while the largest contributors within the top five commodities plus others in each category are depicted in the outer circle (**a**). The per-capita FW are illustrated by 11 aggregated commodities (**b**) and by three waste types; excessive preparation, disposal, and leftovers (**c**).

supply chain. The 28 food items are detailed using the available sectors in NSFIE and are comprised of 197 food commodities in FIES. Importantly, "meats" which generated <3% of the total FW (only 6.7% of the FW of "vegetables") induced almost 10% of the total FWGHG.

### Impact of demographic aging on food waste and the related GHG emissions

Figure 4 depicts the projected results of the impact of future demographic trends between 2015 and 2040 on household FW and FWGHG. The total FW will slightly increase until 2020, by 0.7% (+0.01 Mt) in 2020 compared to the base year (2015) of FW as the total number of households and the total population changes[39]. The total FWGHG will rise until 2019. The total number of households is expected to increase up to 2025, and then decrease until 2040, although in 2030 it is expected that the number of households will be similar to 2015 levels (0.2% higher in 2030 and 4.8% lower by 2040). The national population will continue to decrease until 2040, with a 13% reduction in 2040 compared to 2015. In response to these demographic trends, the total FW and FWGHG in 2040 are estimated to be 5.3% and 6.2% smaller than 2015 levels, respectively. The reason for the different trends in household FW despite decreases in the total number of households and national population is due to the higher ratio of elderly households. The shares of FW and FWGHG by households in their 60 s and those in their 70 s and older will respectively grow between 2015 and 2040. Increases in FW and FWGHG by those in their 70 s and older were markedly higher, by 18% in 2040 compared to 2015 while

reductions achieved markedly by those in their 40 s were 29% during the same period.

In line with the above changes due to the aging and shrinking population in Japan, FW and FWGHG for all food groups (28 groups) will drop. The driver that was the most difficult to reduce within household FW and FWGHG was fresh fruits (−1.2%), followed by other processed vegetables and seaweed (−2.5%) and salted and dried seafood (−2.5%). The most remarkable items, other soft drinks and instant staple foods (e.g., frozen pasta), were estimated to decrease FW and FWGHG by 10.4% and 10.2% from 2015–2040, respectively.

### Discussion

This study elucidates the detailed structures of household FW (not including those related to eating out) and FWGHG for Japan using a combination of socioeconomic statistics for household food consumption and FW alongside LCA. The major drivers of FW for Japan were attributed to vegetables and fruits, consistent with a previous study[30]. Although it is difficult to compare previous studies directly due to differing definitions of FW and quantification methods, a few studies estimated the average per-capita FW of households at the national level, such as in Germany, the US, Finland, and China at roughly 60-100[40], 124[41], 23[42], and 39 kg/cap·yr[13], respectively. The household FW quantified in this study among age brackets ranged from 16.6−46.0 kg/cap·yr, a seemingly reasonable amount of per-capita FW.

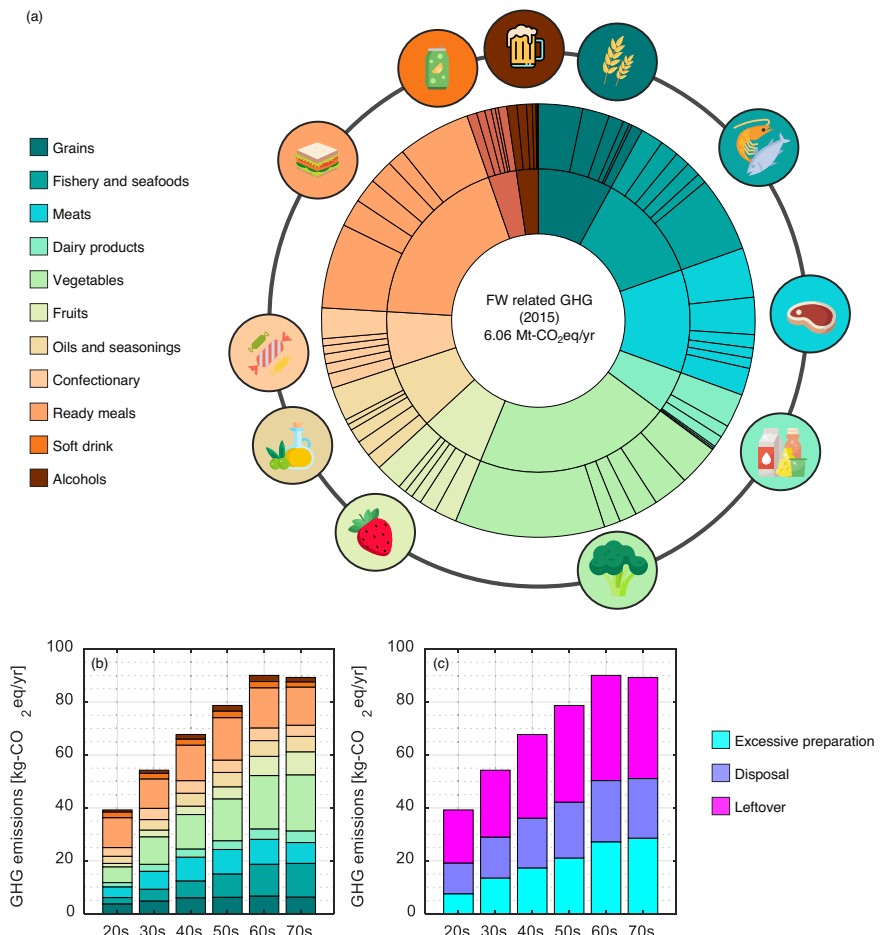

**Fig. 2 | Compositions of household FW-related GHG emissions (FWGHG) for Japan in 2015.** The total FWGHG is shown by 11 aggregated categories in the inner circle while the largest contributors within the top five commodities plus others in each category are depicted in the outer circle (**a**). The per-capita FWGHG are illustrated by 11 aggregated commodities (**b**) and by three waste types (**c**).

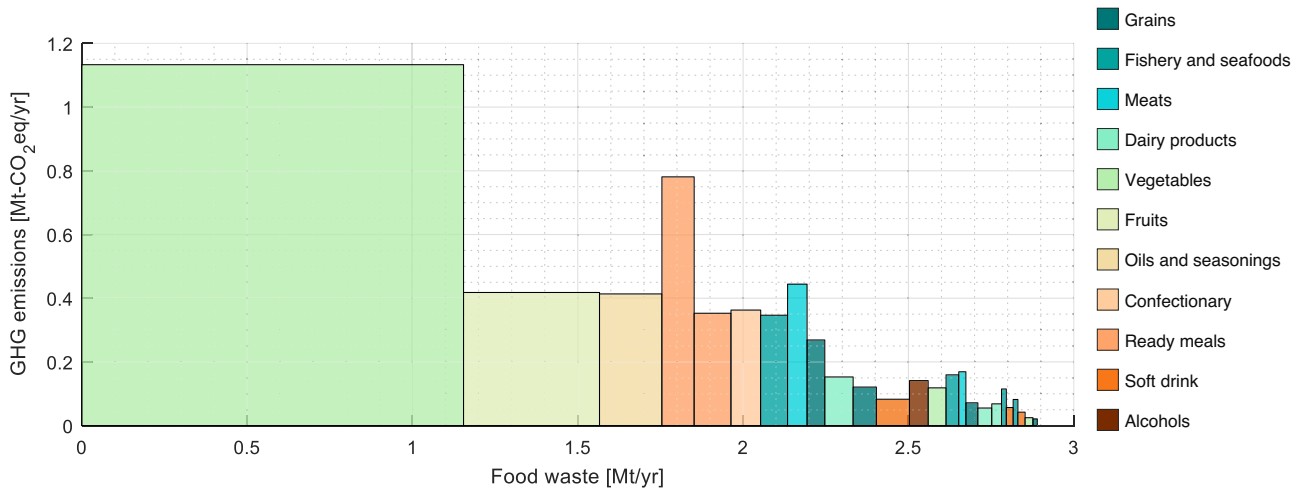

**Fig. 3 | Skyline chart depicting FW (X-axis) and FWGHG (Y-axis) in 2015 detailed according to 28 food groups within 11 categories.** The boxes are arranged in order of the area of the boxes.

The most impactful finding presented by this study is that as households age the average per-capita FW would increase due to differing dietary preferences and purchasing patterns among age brackets of household in Japan. This finding is partially consistent with other domestic statistics which suggested that household FW increases along with the age of the main cooker in the household[34]. The main reason for this increase is that elderly households were more likely to purchase fresh vegetables and fruits compared with other households. These two food types tend to lead to household FW when cooked at home (i.e., via excessive preparation) but are also

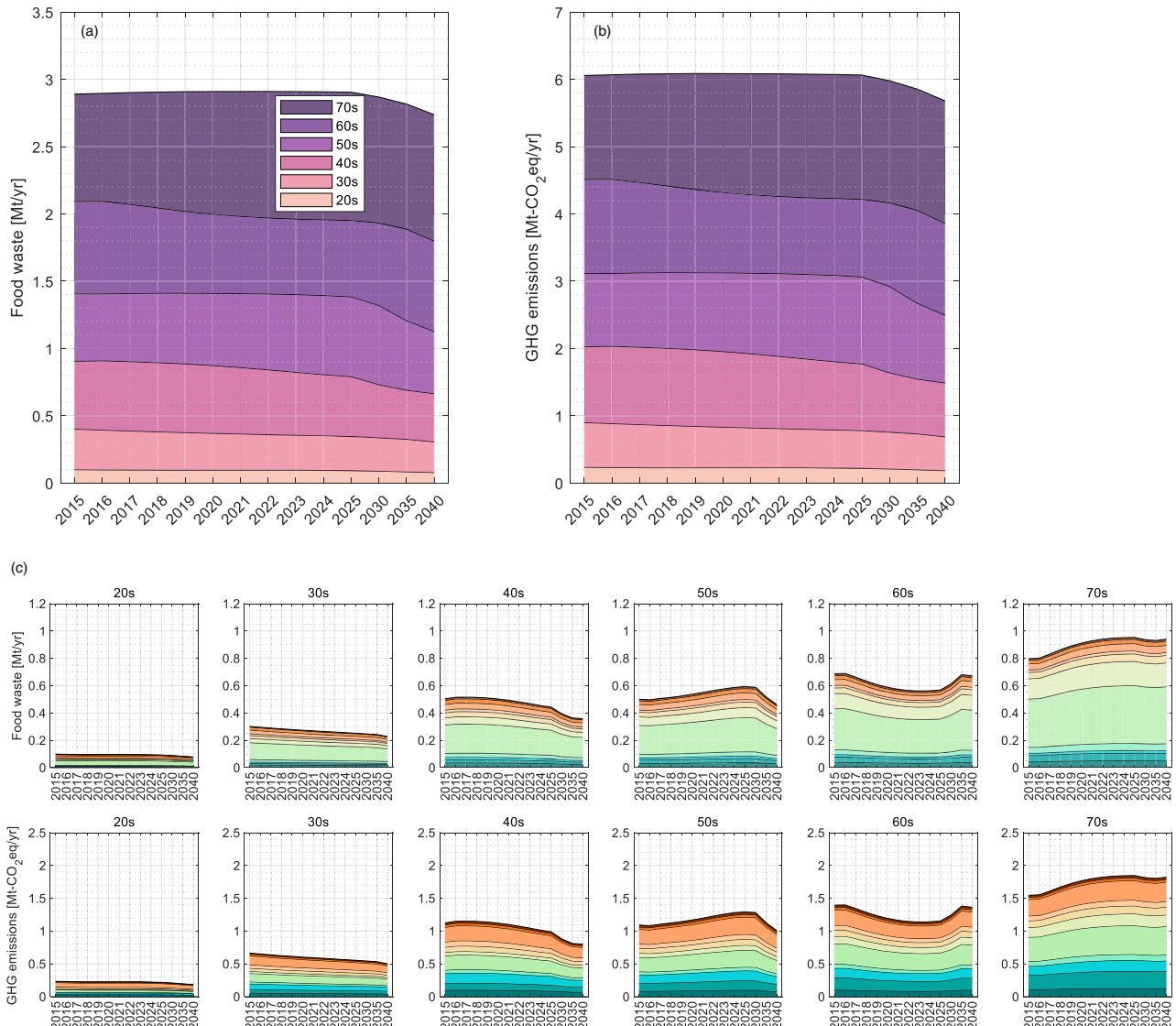

**Fig. 4 | Trends in predicted total household FW and FWGHG from 2015 to 2040 based on demographic change.** Predicted FW (**a**) and FWGHG (**b**) are depicted by age bracket, reflecting food consumption changes associated with trends in the number of households and household size (i.e., population) among the household brackets. The FW and FWGHG by 11 consumption categories during the same period are illustrated among age brackets (**c**).

difficult to preserve for long periods (contributing to direct disposal). On the other hand, according to FIES, elderly households did not go out to eat at restaurants or purchase ready meals as frequently as younger households. The results of FW would be affected if those related to eating out are also considered as household FW; however, it is out of the scope of this study due to the data limitation. In addition, the results of estimating the impact of future demographic trends in this study imply that household FW cannot be expected to decrease in line with the total population and the number of households because of the aging population trend in Japan. As mentioned in the previous section, household FW needs to be reduced to 2.16 Mt by 2030 in order to meet the reduction target commitment by the government. From the base year of this study, 0.73 Mt reduction in household FW is required. According to the estimation results of the demographic impact showing that household FW will slightly decrease by 2030 compared to 2015 (−0.8%, 0.02 Mt), drastic measures are required to reach the FW reduction target.

Regarding these findings, an establishment of policy instruments that are "fit for purpose" rather than a "one size fits all"

approach is clearly needed, particularly for elderly households who are likely to generate the most FW. For example, encouraging them to take a recurrent education program about household FW and the related environmental and social issues is important. From the producer side, it is essential to develop meal kits suitable for this age bracket which are easily cooked without any excessive preparation of food and that can be preserved for a long period in frozen form. These measures are critical not only for the senior generation; however, focusing educational content on this age group could be more effective for suppression of household FW (e.g., it is easier to focus on a specific group if meal-kit menus are developed in line with their dietary preferences and appropriate portion sizes). This sort of consideration can be applied to other households, too. Although younger households generate less household FW than elderly households, they tend to purchase more confectionary, and ready meals, and prefer to dine out instead of cooking at home. Hence, it is necessary to consider a set of dietary and customized policy measurements cognizant of both health risks and FW. Further, it is imperative to inform all households about how to make effective use of food materials, particularly vegetables

and fruits which are more likely to generate household FW, including recipes that promote smart use of foods at differing levels of freshness.

The national FWGHG quantified in this study was around 6.06 Mt-$CO_2$eq/yr, which accounted for almost 20% of the life cycle GHG emissions of households associated with food (excluding dining out). This number is close to the result of FWGHG (6.80 Mt-$CO_2$eq/yr) shown by a previous study[30] applying an input-output life cycle inventory methodology[43]. The per-capita average FWGHG by age bracket quantified ranged between 39.2 and 90.1 kg-$CO_2$eq/yr. This is around 1% of the per-capita household GHG emissions for all commodities in 2015[44], implying that the reduction potential of GHG emissions through a reduction in household FW is not likely to be effective in Japan. This implication is consistent with a previous study conducting a meta-analysis quantifying the potential reductions in household life cycle GHG emissions (i.e., carbon footprints) via consumer choices using an environmental input-output approach[45]. Nevertheless, it is desirable to reduce unnecessary GHG emissions related to consumption. Toward this end, dealing with meat products plays a crucial role in reducing FWGHG. FW of "meats" is >15 times smaller than the FW of "vegetables", contributing just 2.8% of the total FW while FWGHG of "meats" accounts for almost 11% of the total FWGHG. In fact, cutting FW for just three items within "meats" (i.e., *beef*, *chicken*, and *pork*) could reduce >7% of total FWGHG. Hence, it would be relatively easy for consumers to suppress FWGHG by eating smaller portions of meat and avoiding waste. To do so, making and distributing *how-tos* for using the wasted parts of meat effectively could be helpful. Other than meat, according to the results, using foods such as milk, mushrooms, tomato, and tofu that heavily impact FWGHG and can be easily damaged could also be prioritized.

Although not detailed here due to being out of scope, how to improve not only environmental outcomes but also human health outcomes through dietary changes is highly relevant[46–49]. For diets in most countries, a decrease in the intake of red and processed meats and dairy products along with an increase in vegetables and fruits is considered to lead to better health outcomes along with a significant reduction in GHG emissions[50–52]. However, the potential trade-off between such a dietary shift and household FW is underpinned by the results obtained in this study. Therefore, recommended diets should be promoted along with education specific to the potential spontaneous increase in household FW as the demand for vegetables and fruits increases.

We examined uncertainties in our analysis with respect to FW ratios and inedible ratios based on the statistics used, and found them to be small enough not to seriously affect the results and discussion (see the Supporting Result and Supporting Discussion). However, since some limitations remain as detailed in the Methodology, there is room for further clarification of household FW and FWGHG in Japan in future studies. While this study focuses on the demand-side strategies for a reduction in FWGHG along with FW, it is essential to analyze details of FWGHG at the different stages of the supply chain using an LCA approach to elucidate supply-side decarbonization strategies. In addition, As recent footprint studies have begun to uncover[53–57], utilizing consumption expenditure microdata will allow for a more comprehensive picture of the regional diversity of household FW in Japan and household socio-economic factors which are statistically associated with this diversity[36,58]. Finally, although the central objective of this study is to highlight the FW and FWGHG compositions among different age brackets and the impact of future demographic trends on them, we bear in mind that a more accurate projection for them needs to consider changes in factors such as food supply chains, other socio-demographic structures (e.g., household income), consumption behavior, and dietary preference shifts[59] (see the Supporting Result and

Supporting Discussion). Incorporating these factors via econometric analysis is a future goal of this study.

## Methods
### Quantification of the household food waste by household age bracket
The methodology utilized in this study can be summarized as follows: (1) we calculated the household FW intensity by combining the food waste statistics and the household- and food-related economic statistics of Japan. (2) The structure of total household and age bracket FW were estimated. (3) The total FWGHG from raw materials to retail and those by age bracket were quantified using life cycle emission inventories and economic and environmental statistics. (4) Based on the future trend of the number of households and predicted family size for each age bracket, the impact of future demographic changes on FW and FWGHG were projected.

For the food waste statistics, we employed the Standard Tables of Food Composition in Japan – 2015 - (Seventh Revised Version; STFC)[60] and the Food Loss Statistics Survey (FLSS)[34]. The STFC has been published seven times since 1950 and the latest version was published in 2020. We chose the tables for 2015 as they were the closest to the analysis year in this study. This allowed us to obtain the average inedible ratios as well as 52 basic nutrients of an edible part of the 2,191 food products. Here, the average inedible ratio was defined as "the portion of the food that is discarded in normal eating habits as a percentage of the whole food in purchased form"[60]. FLSS provides the physical amount of food waste for food categories generated from Japanese households based on a survey utilizing actual measurements and recording, allowing for the obtainment of average ratios of food waste compared to the edible portion of foods consumed in households. The survey target households' were selected from a sampling list prepared by collecting information from local governments and other organizations and by public solicitation. The survey was conducted through the MAFF (Ministry of Agriculture Forestry and Fisheries, Japan)-Private Sector flow, and was conducted using the self-reporting method, in which survey forms were distributed to and collected from surveyed households[34].

For the household- and food-related economic statistics, the Family Income Expenditure Survey (FIES) describes the average consumption expenditures for household attributes, such as income level, family size, and age of the householder (i.e., the highest income earner). Expenditures for ~200 food items are also available. There are gaps in terms of foods and units (i.e., physical vs monetary) between these three statistical sources. To account for this gap, it is necessary to convert monetary amounts from the FIES into physical amounts consumed per household. To achieve this, we employed the Retail Price Survey (RPS)[61] to carry out the conversion for consumption expenditure from the FIES. The RPS allows for the estimation of physical food consumption per unit price of each item (i.e., kg/JPY). The physical consumption per unit of price for some food items in the RPS however is not provided in kg/JPY (e.g., 149 JPY/piece for instant noodles). In these cases, we chose a representative commodity classified within the food item category to determine the physical amount per unit price.

Through the integration of the above statistics and procedures, we obtained the average FW per expenditure, $w_{ik}$, using Eq. (1).

$$w_{ik} = u_i q_i \sigma_i r_{ik} \tag{1}$$

where $u_i$ and $q_i$ denote the physical unit per unit of price (e.g., 1 package/100 JPY) and the corresponding weight per physical unit (e.g., 100 g/1 package) for the food item retrieved from RPS. $\sigma_i = (1 - \rho_i)$ denotes the survival ratio which was calculated by deducting one from the inedible ratio $\rho_i$ retrieved from STFC. We describe how to obtain $\sigma_i$ from $\rho_i$ in the Supplementary Information (SI). $r_{ik}$ denotes the FW ratio by type (i.e., the amount of FW per food consumed per household) retrieved from FLSS. $i$ and $k$ represent each food commodity from FIES

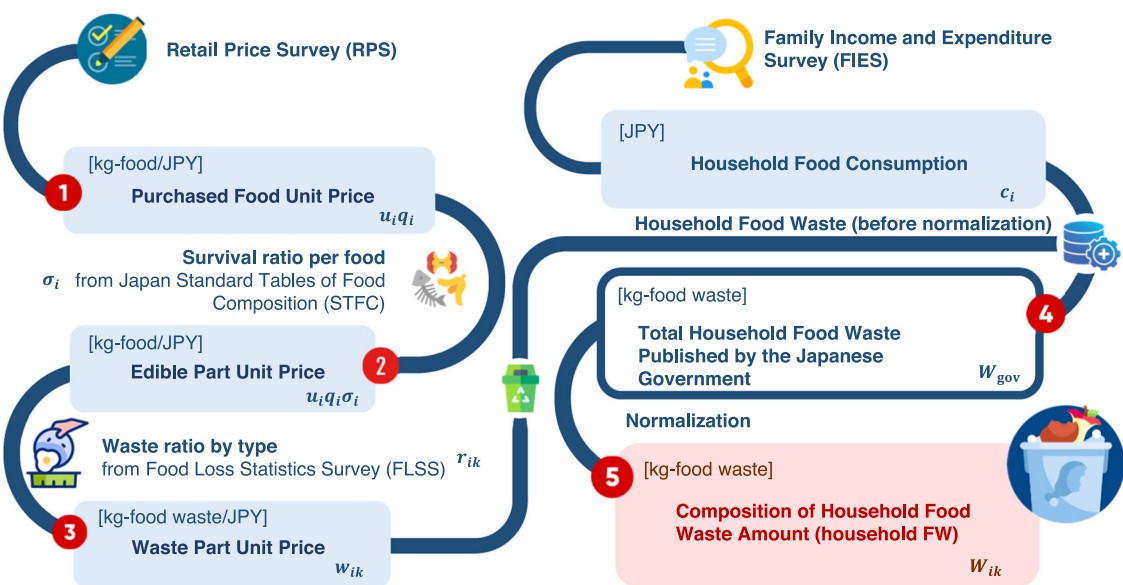

**Fig. 5 | Schematic diagram of the methodology of this study to quantify the structure of household FW in Japan.** The variables (in bold italic) are presented in the equations in the main text in Eqs. (1) and (2).

and waste type (i.e., "excess preparation", "disposal" and "leftovers") from the FLSS, respectively. To utilize Eq. (1), we first prepared concordance tables for mapping the food items in FLSS and RPS onto FIES. For some food items that are not covered by RPS, we found the representative food and determined the retail price per unit amount based on the price and amounts retrieved from official producer websites (Meiji, Nissui, Morinaga, Kikkoman, and CHOYA) and one of the largest retailers in Japan (Topvalue). The concordance table is shown in Supplementary Data 1. Using $w_{ik}$ and $W_{\text{gov}}$ which is from the reference amount of household FW published by the Japanese government, the household FW on commodity $i$ by waste type $k$, $W_{ik}$, was estimated as shown in Eq. (2).

$$W_{ik} = W_{\text{gov}} \frac{w_{ik} c_i}{\sum_i \sum_k w_{ik} c_i} \qquad (2)$$

where $c_i$ denotes the average per-household annual consumption expenditure on the food item. The need for Eq. (2) arises from the issue that even if the annual total food waste is estimated by multiplying the average per-capita food waste per day on FLSS by 365 (the total number of days in a year) and the total national population, it will not be consistent with $W_{\text{gov}}$ due to the difference in the methodology of accounting used for FW between FLSS and these statistics. Thus, in Eq. (2), $\frac{w_{ik} c_i}{\sum_i \sum_k w_{ik} c_i}$ represents the ratio of household FW on commodity $i$ by waste type $k$ to the total. The methodological flow is illustrated in Fig. 5.

Next, the per-capita household FW by waste type, $\overline{W}_{ik}^b$ is determined in a similar manner to that used to determine $W_{ik}$, as detailed in Eq. (3).

$$\overline{W}_{ik}^b = W_{\text{gov}} \frac{w_{ik} c_i^b}{\sum_b \sum_i \sum_k w_{ik} c_i^b} \times \frac{1}{H^b \sqrt{p^b}} \qquad (3)$$

where $b$ represents the age of the householder ($b = 1$: 29 and younger (20 s), 2: 30–39 (30 s), 3: 40-49 (40 s). 4: 50–59 (50 s), 5: 60–69 (60 s), 6: 70 and older (70 s)) as the household attribute. This study adopted the National Survey on Family Income and Expenditure (NSFIE) instead of FIES for the identification of the generational trends in household FW and FWGHG because the NSFIE allows for utilizing consumption

expenditures for each age group while FIES does not. It should be also noted that because NSFIE records consumption expenditures in a similar manner to FIES, food item $i$ on FIES can be attributed to food group $j$ on NSFIE. We therefore assumed that the purchase share of food items in the attributed group is the same among age brackets and determined $c_i^b = c_j^b \times \frac{c_i}{\sum_i c_{v i \in j}}$ which denotes the per-household annual consumption expenditure by attribute on the food item. $H^b$ represents the number of households by age bracket. $p^b$ denotes the family sizes by attribute. As multiplying the number of family sizes among household attributes directly retrieved from NSFIE by the corresponding number of households does not yield the total population recorded in the national population statistics, we estimated $p^b$ through an optimization procedure as explained in the next subsection. Here the OECD square root scale is employed for the equivalization of consumption expenditure with respect to economies of scale due to different family components[62]. Note that the detailed results of household FW and FWGHG by age bracket were represented based on the number of food groups (28) to avoid the bias engendered by the same purchase share of the food items mentioned detailed. This number is smaller than the total number of food items (197) in the FIES, however, FLSS only provides the average food waste ratio for 21 categories.

## Quantification of lifecycle GHG emissions related to food waste by household bracket

To quantify the impact of estimated household FW on the environment, LCA was applied as in previous studies (see[63]). In this study, the Japanese life cycle inventory database, IDEA v3.1.0[64] was employed to ensure that the LCA procedure was in line with ISO14040. IDEA records the unit of life cycle impacts for about 4700 items based on the Life-cycle Impact assessment Method based on Endpoint modeling (LIME), which is the Japanese life cycle inventory database with the highest resolution currently available. This study employed the IPCC 2013 global warming potential 100a for GHG ($CO_2$, $CH_4$, $N_2O$, HFCs, PFCs, $SF_6$, and $NF_3$) as a midpoint indicator of life cycle impact for expressing the FWGHG. Owing to the richness of the inventory in IDEA, most impacts can be attributed to the food item for food waste (i.e., commodity $i$) in this study, one by one. If multiple food items of IDEA are attributed to the food waste $i$, the arithmetic mean was taken for obtaining the unit life cycle impact. Supplementary Fig. 1 illustrates the system boundary of FWGHG in this study.

It should be noted that the life cycle inventory in IDEA covers from raw materials to the wholesale stage of the food commodity for raw agricultural, meat (except chicken), and fishery products (not including processed foods) while the amount of the corresponding food estimated in this study represents at the retail stage because both the FIES and RPS are based on the purchaser price at the retailer. Hence, we considered GHG emission from the wholesale to the retail stage of these food commodities (including the food processing process; e.g., from boneless cuts of meat to dressed meat) and the per-capita FWGHG by age bracket, $\overline{F}_i^b$, is quantified in Eq. (4).

$$\overline{F}_i^b = \overline{W}_i^b f_i (1+\beta_i) \qquad (4)$$

where $f_i$ denotes the life cycle GHG emission intensity of commodity $i$, retrieved from IDEA. $\beta_i$ denotes the ratio of emissions from production to the wholesale stages and then those from the transport to retail stages which are calculated using 3EID[43]. $\beta_i$ takes the value zero when $i$ is not correspondent to the agricultural, meat, or seafood products (see Supplementary Data 1). Furthermore, the GHG emission intensities of *beef*, *pork*, and *other raw meats* (except *chicken*) in IDEA focus on these boneless meats that are trimmed to become dressed meats at the retail stage (i.e., meat sinews and excess fat are discarded). In response to this, we modified $f_i$ by adopting the reciprocal yield ratio at the wholesale stage, $\alpha_i$; $f_i' = \frac{f_i}{\alpha_i}$ ($i \in beef$, *pork*, and *other raw meats*) under the assumption that the FWGHG should be allocated to the edible part of the food. A detailed explanation of this procedure is outlined in the Supplementary Information.

The inventory is represented by an impact per physical unit (e.g., t-CO$_2$eq/kg) for most of the food commodities, however, it is represented using monetary units (e.g., t-CO$_2$eq/JPY) for products such as ready meals that contained a broad picture of various foods. For these, we transformed the life cycle GHG emission intensity per monetary unit ($\hat{f}_i$) to that of physical units using the corresponding consumption expenditure and weight as $f_i = \frac{\hat{f}_i}{u_i q_i}$. The hat (^) included here indicates the intensity of the monetary unit.

## Projection of the impact of demographic trends on food waste and related GHG emissions

We attempted to project the future trend in household FW and FWGHG along with demographic changes under the assumption that food consumption behaviors among age brackets and the production technologies were fixed from the base year, as presented in Eq. (5).

$$W_i(t) = \sum_b \left( \frac{W_i^b}{H^b} \times \left( \frac{p(t)^b}{p^b} \right)^{\frac{1}{2}} \times H(t)^b \right) \qquad (5)$$

where $W_i(t)$ denotes the total food waste by attribute for the target year. $H(t)^b$ represents the future number of households by age bracket, retrieved from IPSS[39]. Owing to data availability, the base year was set to 2015 and the target years for prediction were 2016, 2017, ..., 2025, 2030, 2035, and 2040 ($t=1$: 2016, ..., 10: 2025: 11: 2030, ..., 13: 2040). $\frac{W_i^b}{H^b}$ indicates the per-household FW in the base year (i.e., 2015). $p(t)^b$ represents the predicted family size by age bracket based on $H(t)^b$ and the total national population which is also retrieved from IPSS (for the retrieval method, see the next subsection). Thus, $\left( \frac{p(t)^b}{p^b} \right)^{\frac{1}{2}}$ implies the effect of changes in the average household sizes associated with demographic change. When predicting the FWGHG in the target year, $F_i(t)$, we replaced $\overline{W}_i^b$ with $\overline{F}_i^b$ in Eq. (5). This simple approach allows for extracting the influence of demographic changes on the analyzed pressures[32,65].

## Ensuring the consistency of population data related to Japanese households

Both the number of households and family sizes that are recorded in household statistics (i.e., FIES and NSFIE) are inconsistent with the total national population statistics (i.e., IPSS). Also, family size by attribute in current and future years is not officially published. This study therefore estimated the family sizes for households in the analyzed years, $\widetilde{p}^b(t)$, to make them consistent with the number of households and the total national population through mathematical programming.

Firstly, the optimized number of family sizes for households in the base year (i.e., 2015), $\widetilde{p}^b$, were calculated using Eqs. (6)–(8).

$$\min_{\widetilde{p}^b} \sum_b \left( \widetilde{p}^b - p^{b,\text{NSFIE}} \right)^2 \qquad (6)$$

s.t.

$$P^{\text{IPSS}} = \sum_b \widetilde{p}^b H^b \qquad (7)$$

$$p^{b,\text{NSFIE}} \le \widetilde{p}^b \qquad (8)$$

where $P^{\text{IPSS}}$ and $H^b$ denote the total national population and the number of households by age bracket in the base year, respectively. These two attributes are retrieved from IPSS. $p^{b,\text{NSFIE}}$ is recorded in NSFIE, denoting the family size by attribute. Equation (7) holds that the total sum of multiplying each family size by the number of households is equal to the national population on IPSS. Equation (8) holds that the predicted family size by age bracket is always larger than the corresponding one retrieved from NSFIE because $\sum_b \widetilde{p}^b H^b$ is smaller than $P^{\text{IPSS}}$.

Analogous to the method for $\widetilde{p}^b$, the optimized number of family sizes for the households during 2016–2040, $\widetilde{p}^b(t)$, were predicted using Eqs. (9)–(11).

$$\min_{\widetilde{p}^b(t)} \sum_b \left( \widetilde{p}^b(t) - \widetilde{p}^b \right)^2 \qquad (9)$$

s.t.

$$P(t)^{\text{IPSS}} = \sum_b \widetilde{p}^b(t) H(t)^b \qquad (10)$$

$$\widetilde{p}^b(t) \le \widetilde{p}^b(t-1) \qquad (11)$$

where $P(t)^{\text{IPSS}}$ and $H(t)^b$ denote the total national population and the number of households by attribute between 2016 and 2040, respectively. Here we assume that no drastic changes in fertility trends occur (e.g., a child boom) during the studied period. Equation (10) states that family size by age bracket in year $t$ will not exceed the corresponding value from the previous year because the national population is predicted to continue decreasing from the base year onwards[39]. Thus, we assume no policy instruments and events causing changes in the Japanese population and lifestyles (e.g.[66–68]) will occur during the analyzed period.

Finally, the optimized $\widetilde{p}^b$ and $\widetilde{p}^b(t)$ values are used by replacing $p^b$ and $p^b(t)$ respectively in Eqs. (3) and (5) in the previous subsections. The optimized family sizes estimated in this study and the number of households by age bracket that are retrieved from IPSS are detailed in Supplementary Table 1.

## Limitations

As with other studies, this study has the following limitations owing to the limited quantitative resources, particularly those for FW, that should be noted. The most important limitation is the discrepancy in the

collection methodologies for household FW from the statistics used in this study. FLSS represents household FW which was collected from a relatively small sample of households (364 households) during 7 days in December. Thus, wasted foods out of this season (i.e., around December) were not taken into account in the survey. This is the reason for inconsistency with the reference amount of FW (i.e., $W_{gov}$) even if the annual total FW is estimated by multiplying the average per-capita FW per day from the FLSS by 365 (the total number of days in a year) and by the total population. Hence, the structure of household FW could be biased. However, the reference FW is also calculated under the assumption that household FW in the municipalities that could not be investigated is the same as the average, as estimated by the survey sample municipalities. Besides, it is not possible to obtain the detailed structure of reference FW. This study, therefore, used the latest data from FLSS that is solely available for the structure of household FW.

Alongside the household FW, the physical amount of food purchased by households would also contain uncertainties. Because RPS records only the average price per unit amount of the food commodity one by one, it is difficult for some commodities which contain a high number of mixed, various foods (e.g., ready meals) to determine a representative commodity. In addition, we applied the current price retrieved from websites for food commodities from the FIES which did not correspond to those in the RPS. This may also engender uncertainty for the quantification of the physical food amount and FW.

For the quantification of the FWGHG, this study employs the IDEA database due to its highly sectoral resolution and basic unit of impact intensities (i.e., t-$CO_2$eq/kg-food). Yet, the differences in production technologies and supply chains between domestic and imported food products were not distinguished, which causes uncertainty. A multi-regional input-output (MRIO) analysis with a global MRIO model[69,70] allows for dealing with differences between domestic and imported products. This study employed IDEA regarding the sectoral resolution and the closeness of the analytical year compared to MRIO models. This is also because it is currently impossible to know which food purchased by each of the households is domestically produced or imported from overseas. For this reason, the difference between yield ratios at the wholesale stage for *beef*, *pork*, and *other raw meats* between domestic and imported meats could not be considered.

There are discrepancies in the analyzed year among statistics and datasets used in this study. Currently, the latest FLSS is for 2014 while FIES and RPS can be used for every year. NSFIE provides the 2014 consumption expenditures for the age brackets of the household. 2015 is the closest year of demographic data by age bracket of household on IPSS and the life cycle inventory data used in this study. Thus, we presumed our household FW and FWGHG are for 2015 and the reference FW is set as 2015[28].

Finally, the calculation approach of this study assumes that the ratios of household FW generated from foods purchased (i.e., $q_{ik}$ in Eq. (1)) are the same among all age brackets. In other words, this study could not deal with the possibility of differences in the behavior of wasting food between households. Future research should address this issue through a detailed investigation of household characteristics as to the ways of generating FW.

### Reporting summary
Further information on research design is available in the Nature Portfolio Reporting Summary linked to this article.

## Data availability
The average household consumption expenditure and those for the age brackets were retrieved from the Family Income and Expenditure Survey (FIES) and the National Survey of Family Income and Expenditure (NSFIE), Japan. The average food prices by food item were retrieved from the Retail Price Survey (RPS). The average food waste ratio from households by food item was retrieved from the Food Loss Statistics Survey (FLSS). These are available from https://www.e-stat.go.jp/. The edible ratio of food in the retail stage was calculated from the Standard Tables of Food Composition in Japan (STFC), which is available from https://www.mext.go.jp/en/policy/science_technology/policy/title01/detail01/1374030.htm. The demographic data (the number of households and the average number of household sizes for the age brackets) were retrieved from the database of the National Institute of Population and Social Security Research (IPSS), which is available from https://www.ipss.go.jp/, and the NSFIE. The life cycle climate impacts (GHG emissions) for the food wastes were created based on the Inventory Database for Environmental Analysis (IDEA) and the Embodied Energy and Emission Intensity Data for Japan Using Input-Output Tables (3EID), which are available from https://sumpo.or.jp/consulting/lca/idea/ and https://www.cger.nies.go.jp/publications/report/d031/eng/index_e.htm, respectively. The data for the analysis and the source of the results except those related to GHG emissions are disclosed in the Supplementary Data and the Source Data, respectively. This is because the GHG emission inventory in IDEA cannot be disseminated license restrictions. Results which enable reverse engineering of IDEA inventory data are also excluded. Source data are provided with this paper.

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

## Acknowledgements
This study was supported by JSPS KAKENHI (grant numbers JP21H03673 and JP24K03149) and Research Institute for Humanity and Nature (RIHN: a constituent member of NIHU) Project No. RIHN14210156. We would like to thank Prof. Shunichi Hienuki for his valuable comments on this study.

## Author contributions
Y.S. proposed the conceptualization of this study. Y.S., A.I., Y.L., A.C. carried out the methodology, visualization, and investigation. Y.S. and A.I. wrote the original draft. Y.S., Y.L. and A.C. reviewed and edited the draft. Y.L. refined the visualization.

## Competing interests
The authors declare no competing interests.
