## [Peer Review File · Nature Communications]

Curbing household food waste and associated climate change impacts in an ageing societyREVIEWER COMMENTS

Reviewer #1 (Remarks to the Author):

Thank you for the opportunity to review the manuscript entitled "Curbing household food waste and associated climate change impacts in an ageing society", submitted for possible publication to Nature Communications. The manuscript deals with an original and topical concern, namely the environmental impacts of household food waste, with specific regards to the six age brackets of Japanese households.

Abstract. The abstract is clear and consistent. I suggest the authors adding few lines, at the beginning, which justify the need to investigate food waste in general and specifically food waste in Japan. I would also add the brief description of the methods adopted to estimate food waste among Japanese households, since such an information is not included in the abstract. Last, I would include some selected quantitative outcomes of the research. Considering that the abstract accounts for 142 words and that the Guide to Authors define that the abstract "should be no more than 200 words", I suppose that the authors have enough space to include the missing information.

Introduction. "Introduction" is clear, consistent and comprehensive, and I have some suggestions/comments to the authors.

L. 38-39. Is it possible to provide a quantitative value together with "one-quarter of total global GHG emissions"?

LL. 39-40. The statistics related to the environmental impacts associated with food waste land-use and land-use changes refers to 2015. Could you please provide updated facts?

LL. 41-43. Could the authors provide statistics related to the global water consumption, as to strengthen the subsequent assumption: "are of high concern"?

L. 48. Is it possible to provide a quantitative value together with "half of global"?

LL. 58-68 are interesting and justify the choice of Japan as a study area. Is it possible to identify the total amount of food waste generated in Japan? I suppose that, if 5.22 Mt were still edible and represented the 22% of the total food waste, about 24 Mt of food waste were generated in Japan. Hence, is it possible to compare such an amount with other international realities, perhaps in terms of food waste per capita, as to highlight the importance of investigate Japanese households on the global scale?

L. 71. "Different consumption patterns segmented by the age group of the household head" seem to represent the originality of the current research. Is it possible to refer to previous similar studies and, in present or not, highlight the originality of the current research? In addition, considering that L. 73-74 defines Japan "as a case study due to its worldwide leading ration of elderly population", is it possible to provide evidence of this? In which percentage are people aged 65 and above present in Japan?

Results. "Results" are presented in a clear manner, both in the Figure 1 and in the main body of the text. The authors have done several efforts to provide consistent results.

In Figure 2, the authors distinguish between the food waste driver, namely "excessive preparation", "disposal" and "leftover". How such information have been segmented? Are these data included in the Family Income Expenditure Survey? Considering that such an aspect is rather blurred in the section "Methodology", I invite the authors to provide more information related to the structure/development/composition of the different surveys used as data sources.

Considering that the research highlights food waste along the supply chain "from cradle to farm or factory (not including emissions related to transport, retail, and cooking)" (LL. 79-80), is it possible to provide a details of the food waste GHG emissions according to the different steps of the supply chain? It could also help understand the most impacting foods, which are wasted among Japanese households, and could help shape strategies towards sustainability and reduction of food waste emissions.

Discussion. "Discussion" are original and interesting. The authors have highlighted similarities and differences with previous studies and have propose possible strategies and policy directions.

Methodology. The description of the methodology is clear and comprehensive. In the next lines, I highlight some comments, which should be addressed by the authors, especially to clarify the environmental impacts assessment process.

LL. 259-263. First, could the authors please provide an overview of the research methodology adopted in the research, highlighting the stepwise approach in terms of food waste measurement, life cycle assessment, etc.? In addition, is it possible to provide some more details related to the "Food Loss Statistics Survey"? How such a survey is conducted in Japan? According to which sampling strategy, to which investigated variables, etc.? In the "Limitations" it results that the survey was conducted among 364 household. Could the authors please provide more information related to the sampling process, the questionnaire development, etc.?

LL. 266-267 could be better clarified. In the field of the sentence: "there are gaps in terms of foods and units", could the authors please develop such a concept further? It should be highlighted the nexus between the information collected through the Food Loss Statistics Survey and those identified in the Family Income Expenditure Survey, by highlighting the variables/questions/dimensions collected in the two different surveys. One graphical representation of the data collection process could be useful.

As a general comment, I suggest the authors distinguishing between two different subsections, namely one addressed to the quantification of household food waste, and the other addressed to the analysis of the life cycle GHG emissions.

LL. 316-317 refer to an authoritative study in the field of the life cycle assessment to estimate the GHG emissions in food waste. Could the authors please define the LCA guidelines adopted in the current research? I suppose that the authors have adopted the guidelines provided in the ISO14040. Hence, I invite the authors to better clarify the stepwise approach, namely: (i) goal and scope definition; (ii) inventory analysis; (iii) impact assessment; and (iv) interpretation of the results. The addition of these information could strengthen the environmental impact assessment analysis. Which software was used to perform calculations? Has an uncertainty/sensitivity analysis been performed?

Reviewer #2 (Remarks to the Author):

Reviewer thinks it is a novel research which estimates the food waste (FW) detailed by age brackets and conducts future estimation. However there still need improvement in the model.

- This model assumes that the consumption in each age bracket does not change through 2015 to 2040 except the effect of household size. This is a strong assumption because consumption will chage by various factors such as cohort effect, trend of the times, price, income.

See The estimation of The Policy Research Institute of the Ministry of Agriculture, Forestry and Fisheries (PRIMAFF)

https://www.maff.go.jp/primaff/seika/attach/pdf/190830_1.pdf

(in Japanese)

-The food waste ratio in FLSS indicate food loss (edible part) per food used (edible part). On the other hands , the weightin FIES and RPS is retail weight which may include inedible part. There seems no factor to fill the weight gaps.

- Inventory data of IDEA for agricultural and fishiery products are based on the weight at wholesale stage. However, weight and physical units used in this study is based on the retail stage. This may lead underestimation of FWGHG.

-The low externality of food results in more food waste from elderly households. However, this can be said to result in less food waste from food services and other sources. Is there any implication to he national target for FW reduction?

To Reviewer 1:

Comment

1. Abstract. The abstract is clear and consistent. I suggest the authors adding few lines, at the beginning, which justify the need to investigate food waste in general and specifically food waste in Japan. I would also add the brief description of the methods adopted to estimate food waste among Japanese households, since such an information is not included in the abstract. Last, I would include some selected quantitative outcomes of the research. Considering that the abstract accounts for 142 words and that the Guide to Authors define that the abstract "should be no more than 200 words", I suppose that the authors have enough space to include the missing information.

Response:

Thank you so much for your kind advice with consideration of word limits. We have added a brief description of the methodology and quantitative outcomes of this study. The revised abstract is as below.

"We unveiled the intricate quantitative structure of household food waste and their corresponding life cycle greenhouse gas emissions from raw materials to retail utilizing a combination of household- and food-related economic statistics and life cycle assessment in Japan. Given Japan's status as a nation heavily impacted by an aging population, this study estimates these indicators for the six age brackets of Japanese households, showing that per capita food waste increases as the age of the household head increases (from 16.5 for the 20's and younger group to 46.0 kg/year for 70's and older in 2015) primarily attributed to the propensity of older households purchase of more fruits and vegetables. Further, the largest life cycle greenhouse gases related to food waste was 93.1 kg-CO₂eq/year for those in their 60's while the smallest was 39.4 kg-CO₂eq/year for 20's and younger. Furthermore, food waste and associated emissions are expected to remain constant due to future demographic changes imparted by an aging, shrinking population until 2040. "Fit for purpose" measures focused on demographic shifts are crucial for Japan and other countries with similar dietary patterns and demographics to achieve related sustainable development goals through suppressing food waste and associated emissions under new dietary regimes."

2. Introduction. "Introduction" is clear, consistent and comprehensive, and I have some suggestions/comments to the authors.

Response:

Thank you for your comments which we address in order below.

3. L. 38-39. Is it possible to provide a quantitative value together with “one-quarter of total global GHG emissions”?

Response:

Thank you for the suggestion. Related to the next comment, we have modified the corresponding sentence to make it more consistent regarding the subsequent sentence. The revised sentence mentions “*one-third of total global GHG emissions, namely 17 giga tons (10⁹ ton: Gt) of carbon dioxide equivalent (CO₂eq)*” according to Crippa et al. (2021).

4. LL. 39-40. The statistics related to the environmental impacts associated with food waste land-use and land-use changes refers to 2015. Could you please provide updated facts?

Response:

To our knowledge, an increasing amount of global GHG emissions associated with the food chain (not limited by food waste as you see in the original sentence) provided by Crippa et al. (2021) is estimated by the EDGAR (Emissions Database for Global Atmospheric Research)-Food which is a global food emissions database. Since the data range had been updated until 2018 when we assessed the database in November 2023, we have rewritten the sentence as follows:

“Using life cycle assessment (LCA), it was found that the direct and indirect greenhouse gas (GHG) emissions associated with the international food chain accounted for more than one-third of total global GHG emissions, namely 17 giga tons (10⁹ ton: Gt) ⁶ of carbon dioxide equivalent (CO₂eq) and these emissions increased by approximately **2.2** Gt-CO₂eq from 1990 to **2018** considering land-use and land-use changes (LULUC) resulting from food production ⁷.”

5. LL. 41-43. Could the authors provide statistics related to the global water consumption, as to strengthen the subsequent assumption: “are of high concern”?

Response:

We referred to “Vanham, D., Mekonnen, M. M. & Hoekstra, A. Y. The water footprint of the EU for different diets. *Ecol. Indic.* **32**, 1–8 (2013)” to support this assumption.

6. L. 48. Is it possible to provide a quantitative value together with “half of global”?

Response:

Thank you for the kind advice. We have included the quantitative value (9.3 Gt-CO₂eq) here.

7. LL. 58-68 are interesting and justify the choice of Japan as a study area. Is it possible to identify the total amount of food waste generated in Japan? I suppose that, if 5.22 Mt were still edible and represented the 22% of the total food waste, about 24 Mt of food waste were generated in Japan. Hence, is it possible to compare such an amount with other international realities, perhaps in terms of food waste per capita, as to highlight the importance of investigate Japanese households on the global scale?

Response:

Thank you for your clarification. We should have written “22% of total food loss and waste” instead of “22% of total food waste” in this sentence. This comes from the different definitions of “food waste” between Japan and the United Nations, sorry for this confusion. Therefore, 5.23 Mt of food waste (i.e., associated with household, food service, and retail) within 2.44 Mt of food “loss and waste” were generated in Japan in 2021 (Consumer Affairs Agency, Government of Japan, 2023). This can be translated into 41.7 kg/cap. Comparable food waste per capita across countries is, however, limited in terms of food waste because of different methods and timelines (UN, 2021). According to the UNEP Food Waste Index Report 2021 (UN, 2021), Japanese food waste per capita was reported to be 64 kg/cap (medium confidence) in 2017. In the same report, the global estimation of food waste denotes 121 kg/cap, which is almost twice as high as for Japan. However, the estimated food waste in economically developed nations such as the United States, New Zealand, Netherlands, and Austria, were reported to be lower than that in Japan, accounting for 59, 61, 50, and 39 kg/cap (high confidence), respectively. Therefore, we believe it is valuable to investigate the relationship between Japanese households and household food waste cognizant of its relatively high amount compared to similarly developed nations.

Based on the above comment, we have edited the main text following our description of the aging issue to include the following text: “In addition, Japan, compared to similarly developed

nations such as the US, the Netherlands, Australia and New Zealand has a relatively high level of food waster per capita (UN, 2021).”

8. L. 71. “Different consumption patterns segmented by the age group of the household head” seem to represent the originality of the current research. Is it possible to refer to previous similar studies and, in present or not, highlight the originality of the current research? In addition, considering that L. 73-74 defines Japan “as a case study due to its worldwide leading ration of elderly population”, is it possible to provide evidence of this? In which percentage are people aged 65 and above present in Japan?

Response:

This comment is very helpful to enhance the novelty of this study. First, many researchers have analyzed the structure of lifecycle GHG emissions induced by household consumption (namely, household carbon footprints) with respect to different consumption patterns segmented by the age group of the household head (e.g., Chapman and Shigetomi, 2018; Long et al., 2018, 2020; Shigetomi et al., 2014, 2015, 2018 for Japan; Kronenberg, 2009 for Germany; Liu and Zhang, 2022 for China, Zhang et al., 2022 for multiple nations). These studies highlight the different characteristics in the life cycle GHG emissions among age groups (e.g., elderly households are more likely to generate emissions related to fishery consumption than young households) (Shigetomi et al., 2014). However, to the best of our knowledge, there is no research focusing on the quantification of household food waste by age group. This point is therefore one of the distinctive novelties of this study.

Next, the percentage of those aged 65 and above can be compared among nations according to the UN World Population Prospect 2022, which shows that Japan’s percentage in 2021 (the latest year in the reference) was the second highest (the highest percentage was shown to be in Monaco) (United Nations Department of Economic and Social Affairs, Population Division, 2022). We have added the latest percentage and the reference in the main text. Thank you for this helpful comment.

9. Results. “Results” are presented in a clear manner, both in the Figure 1 and in the main body of the text. The authors have done several efforts to provide consistent results.

Response:

Thank you.

10. In Figure 2, the authors distinguish between the food waste driver, namely “excessive preparation”, “disposal” and “leftover”. How such information have been segmented? Are these data included in the Family Income Expenditure Survey? Considering that such an aspect is rather blurred in the section “Methodology”, I invite the authors to provide more information related to the structure/development/composition of the different surveys used as data sources.

Response:

Thank you for this important insight. These three kinds of food waste data (“excessive preparation”, “disposal”, and “leftover”) are included in the Food Loss Statistics Survey (FLSS), not in the Family Income Expenditure Survey (FIES). FLSS reports both the food-use amount (only edible part) and the average food waste amounts for those kinds (g-food per day) by food category per day. For example, the survey represents that in general meat and meat products are consumed at 55 g per day and those wasted by excessive preparation (too much cutting), disposal, and leftover at 0.7, 0.5, and 0.2 g, respectively. These are based on the actual measurement values taken by the households collaborating with FLSS. We determined the food waste ratio by kind of waste (γ_{ik}) based on the information from FLSS. The numbers related to food consumption and associated food wastes are created by 364 samples, which is one of the limitations of this study, however (as already detailed in the Limitations section). We have complemented this point in the response to your comment #14 as it asks about the details of FLSS in terms of the sampling method.

Conclusively, we have augmented this response alongside our response to comment #14 to explain the statistics used in this study more clearly in the Methodology section in terms of the structure, development, and composition.

11. Considering that the research highlights food waste along the supply chain “from cradle to farm or factory (not including emissions related to transport, retail, and cooking)” (LL. 79-80), is it possible to provide a details of the food waste GHG emissions according to the different steps of the supply chain? It could also help understand the most impacting foods, which are wasted among Japanese households, and could help shape strategies towards sustainability and reduction of food waste emissions.

Response:

Thank you for your insightful comments and suggestions regarding our research. You raised an important point about providing detailed information on carbon emissions from food waste at different stages of the supply chain, particularly in the context of households. This aspect indeed holds significant potential for shaping strategies toward sustainability and reducing FWGHG emissions. To address this gap, we explored previous relevant studies and found that the environmental input-output analysis (EIOA) approach could yield the results you mentioned (For example, see Tantiwatthanaphanich et al., 2022). However, in our study, we utilized a combined method with LCA (IDEA) and EIOA (3EID), and focused on household food consumption behavior, which, as you rightly noted, does not support the granular view of food waste details at each stage of the supply chain. IDEA solely provides the inventory of GHG emissions accumulated through the life cycle of (food) commodity, allowing us to quantify the total emissions resultant from the consumption from cradle to the retailer (for this, please also see our reply to your comment #14). It is necessary to utilize LCA software (e.g., MILCA; <https://www.milca-milca.net/english/index.php>) to break down the emission inventory and analyze the FWGHG at each stage of the supply chain.

Our study aims to highlight the significance of household contributions in mitigating climate impact. We posit that curbing emissions associated with FW at various stages of the supply chain is akin to reducing emissions from the entire food production process, irrespective of waste. This is because producers, when applying decarbonization measures in production, cannot selectively target emissions from specific portions of food. For instance, from a production perspective, if only half of a fish is consumed and the other half is wasted, the emissions associated with the wasted half are proportionally the same as those for the consumed half. The emissions do not change whether the fish is fully consumed or partially wasted. Therefore, this question is similar to food supply-chain emission reduction, which has been well-studied in recent years (e.g., Kummu et al., 2012; Crippa et al., 2021). In our research, we found the behavior of households in food terms of consumption plays a pivotal role in diminishing FWGHG, and focused on the variations in household FW and the challenges posed by an aging population.

While our current study does not incorporate emissions associated with FW from a supply-chain perspective, we believe this topic is crucial for subsequent discussions on decarbonization from the producer's side. We greatly appreciate your suggestion and find this particular point of inquiry extremely interesting. We have included the above discussion in our paper, which can be referred to in the section beginning at Line 334.

12. Discussion. "Discussion" are original and interesting. The authors have highlighted

similarities and differences with previous studies and have propose possible strategies and policy directions.

Response:

Thank you!

13. Methodology. The description of the methodology is clear and comprehensive. In the next lines, I highlight some comments, which should be addressed by the authors, especially to clarify the environmental impacts assessment process.

Response:

Thank you. We have addressed your concerns by clarifying the structures of the database and statistics in our following responses.

14. LL. 259-263. First, could the authors please provide an overview of the research methodology adopted in the research, highlighting the stepwise approach in terms of food waste measurement, life cycle assessment, etc.? In addition, is it possible to provide some more details related to the “Food Loss Statistics Survey”? How such a survey is conducted in Japan? According to which sampling strategy, to which investigated variables, etc.? In the “Limitations” it results that the survey was conducted among 364 household. Could the authors please provide more information related to the sampling process, the questionnaire development, etc.?

Response:

Thank you for these clarifications. The overview of the methodology is as follows: (1) we attempted to calculate the household FW intensity by food by combining the surveys related to food waste (i.e., FLSS and the Standard Tables of Food Composition in Japan (STFC), which was newly introduced to improve the analysis of this study based on other reviewer’s comments) and consumption expenditure (Family Income and Expenditure Survey (FIES) and Retail Price Survey (RPS)). (2) By using the intensity, FIES, and NSFIE, the structures of household FW and those by age bracket were estimated. (3) the total FWGHG from raw material to the retailer (due to the modification in response to the other reviewer’s comment #4) and those by age bracket were quantified by using the IDEA database and an environmental input-output inventory (3EID). (4) Based on the future trend in the number of households and predicted

family sizes for each age bracket, the impact of the future demographic changes on FW and FWGHG was projected.

To improve the readability of the methodology, we have added these explanations at the top of the methodology section. In addition, we have split the section for quantification of household FW and FWGHG into two for each of the methodologies to represent the methodology stepwisely. Further, we have added schematic figures to represent the flow of calculating the FW (this relates to our reply to the next comment) and the system boundary of the FWGHG (our reply to your comment #17), respectively.

Next, the comment requesting the additional explanation of FLSS is related to the above Comment #10. We have complemented the details of FLSS in particular in light of the sampling processes and questionnaire development by referring to the survey reference (MAFF, 2016; https://www.maff.go.jp/j/tokei/kouhyou/syokuhin_loss/gaiyou/index.html#2) as follows: FLSS has been conducted as a general statistical survey by the Ministry of Agriculture, Forestry and Fisheries Japan until 2014. The survey targets representing households were selected from a sampling list prepared by collecting information from local governments and other organizations and by public solicitation. The latest sampling period was December 2014, which consisted of seven consecutive days (one week). The survey was conducted through the MAFF-Private Sector flow and was conducted using the self-reporting method, in which survey forms were distributed to and collected from surveyed households. The survey forms were filled out by the surveyed households through actual measurement and recording (MAFF, 2016). Due to the survey method, the sample sizes are limited, accounting for 364. As far as we know, however, this survey result for 2014 provides the most detailed compositions of household food waste in Japan. In addition, FLSS allows us to retrieve both the food-use amount (only edible part) and the average food waste amounts for those kinds (g-food per day) by food category per day (as explained in response to your comment #10). Therefore, this study used data from the FLSS in this study. The additional explanation of FLSS has been described in the methodology section from L. 260.

15. LL. 266-267 could be better clarified. In the field of the sentence: “there are gaps in terms of foods and units”, could the authors please develop such a concept further? It should be highlighted the nexus between the information collected through the Food Loss Statistics Survey and those identified in the Family Income Expenditure Survey, by highlighting the variables/questions/dimensions collected in the two different surveys. One graphical representation of the data collection process could be useful.

Response:

We agree that providing this sort of process flow is very important for the clarification of the study’s method. Since this comment is related to other reviewer’s comments, we needed to update the methodology to fill the gap of weights between the edible parts and edible plus non-edible parts of food (see also Reply to Comment 3 from Reviewer 2). Based on these amendments, we have added a schematic figure to represent how the different surveys were combined to quantify household FW as shown in Fig. 4 in the revised manuscript (here, Fig. R1).

Fig. R1. Schematic diagram of the methodology of this study to quantify the structure of household FW in Japan. The variables (in bold italic) are presented in the equations in the main text from equations (1) to (2) in the revised manuscript.

Consequently, even after this update, the main discussion and conclusion of this study have not been changed because the amount of household FW was calibrated in line with the total amount that is published by MAFF in our methodology. This change is also reflected in the figure in the main text.

- As a general comment, I suggest the authors distinguishing between two different subsections, namely one addressed to the quantification of household food waste, and the other addressed to the analysis of the life cycle GHG emissions.

Response:

Thank you for the constructive comment. We have reorganized the results section, presenting the household FW and the lifecycle GHG emissions (i.e., FWGHG in this manuscript) in discrete subsections. This reorganization will aid the reader and has improved our manuscript. Thank you for your kind guidance here.

17. LL. 316-317 refer to an authoritative study in the field of the life cycle assessment to estimate the GHG emissions in food waste. Could the authors please define the LCA guidelines adopted in the current research? I suppose that the authors have adopted the guidelines provided in the ISO14040. Hence, I invite the authors to better clarify the stepwise approach, namely: (i) goal and scope definition; (ii) inventory analysis; (iii) impact assessment; and (iv) interpretation of the results. The addition of these information could strengthen the environmental impact assessment analysis. Which software was used to perform calculations? Has an uncertainty/sensitivity analysis been performed?

Response:

This comment is essential for confirming the methodological robustness. We used IDEA version 3.1 (AIST, 2021) which provides the Japanese life cycle impact inventory of about 1,800 products. The data point of GHG emissions in this study is the midpoint as an impact assessment (i.e., GWP in IPCC2013). Hence, both (ii) inventory analysis and (iii) impact assessment in line with ISO14040 are also assured by IDEA. The software that we used for the calculation included Microsoft Excel and MATLAB.

For (i) goal and scope definition, the system boundary is from raw materials to wholesale trade covered in IDEA; for example, GHG emissions generated by producing, packaging, and preserving the food of interest. To enhance this, we have added a schematic figure in the SI as Supplementary Fig. 1 (here, Fig. R2).

Fig. R2. Food life cycle and the system boundary of the FWGHG in this study.

In addition, we recognized to consider that the system boundary of the emission inventory should be modified to meet the food waste amount quantified because the amounts of agricultural and fishery products were estimated based on FIES (or NSFIE) and RPS both relating to a *purchaser price* (i.e., *producer price* + margin cost), as indicated by the other reviewer (see Comment 4 from Reviewer 2). We, therefore, attempted to update the life cycle GHG emission intensities for agricultural and fishery products on IDEA to those from raw materials to retail trade by using the Embodied Energy and Emission Intensity Data for Japan Using Input-Output Tables (3EID) (Nansai et al., 2020). This provides the direct GHG emission coefficients for commodities based on Japan’s input-output table (JIOT). Combining JIOT and 3EID allows us to calculate the breakdown of emissions from the production to the retailer (i.e., emissions related to the margins). Since it is extremely difficult to transform the food weights at the retail stage that were estimated in this study to those at the wholesale stage, we calculated the emission ratio of the total emissions from production to retail stages to those from production to wholesale stages and multiplied it by the corresponding emission intensity (intensities) on IDEA. This procedure aims to complement the emission intensity on IDEA from wholesale to retail stages. To do so, we have modified Equation (4) in the original text as follows for the case of quantifying FWGHG for raw agricultural and fishery products. The per-capita FWGHG by age bracket, \bar{F}_i^b , is quantified in Equation (R1). Then, $i \in$ raw agricultural and fishery products.

$$\bar{F}_i^b = \bar{W}_i^b f_i (1 + \beta_i) \quad (R1)$$

where f_i denotes the life cycle GHG emission intensity from raw material to wholesale on

commodity i (i.e., IDEA). β_i denotes the ratio of emissions from production to wholesale stages to those from transports to retail stages that can be calculated by 3EID.

Regarding (iv) interpretation of the results, it is essential to represent the result of an uncertainty/sensitivity analysis. If the distributions of per-household food consumption, waste patterns, prices of food commodities, and the ranges of the life cycle inventory of food can be obtained, it is possible to conduct a detailed sensitivity analysis. However, these (micro) data are not available. Therefore, we considered the FW ratio and the inedible ratio as a sensitivity factor and conducted the Monte Carlo simulations ($N=1,000$) to estimate the variations and 95% confidence intervals (95%CI) of FW and FWGHG with respect to each of them, respectively.

(A) FW ratio based on FLSS

FLSS also presents the household FW compositions per food consumed by single (36), two-person (168), and more than three-person households (142), respectively (these numbers in the parenthesis denote the number of the sample households). Those food waste compositions recorded in FLSS imply that the FW ratios per category among the household brackets do not necessarily seem to be correlated with the family size. Hence, we assumed the possible FW ratios for each food category would vary between the lowest and highest ratios that can be calculated by summations of the maximum and minimum ratios of FW by kind of waste.

The result presents the total FW and FWGHG accounted for 2.89 ± 0.01 Mt/yr and 6.40 ± 0.03 Mt-CO₂eq/yr, respectively. The reason why the total FWGHG were larger than that of the main result (6.10 Mt-CO₂eq/yr) is that the FW composition was changed from the main result (i.e., when using the mean FW ratios across all of the sample households on FLSS). For the food commodities, Most of them showed 0~2% changes in their FW and FWGHG, and some related to ready meals represented around 3%. However, these changes are trivial enough to keep the discussion regarding the importance of food for FW and FWGHG that we have already described in the main text. Fig. R3 represents the ranges of predicted FW and FWGHG for 11 food categories, showing the sensitivities. This figure is also included in the SI.

Fig. R3. Comparisons of household FW (a) and FWGHG (b) for the three different waste ratios across food categories. “Mean” denotes the result of the mean FW ratios across the three age brackets. “Upper bound” and “Lower bound” denote the cases in which the FW ratios were taken between the maximum and minimum values across the three age brackets on the FLSS, respectively.

(B) Inedible ratio based on STFC

As we replied to your comment #14, We newly referred to the database on the Standard Tables of Food Composition in Japan - 2015 - (Seventh Revised Version) published by the Ministry of Education, Culture, Sports, Science and Technology, Japan (MEXT) in order to fill the weight gap between the statistics used in this study as indicated by the other reviewer. Specifically, we confirmed that FLSS indicates the intake and waste of *edible* food while our estimation of the amount of food consumption by household attribute focuses on a portion of a whole food (*edible + inedible*) by reviewing the document. STFC has been published seven times since 1950 and the latest version is for 2020. We chose the tables for 2015 which is the closest to the analysis year in this study. It allows us to obtain the average refuse rate as well as 52 basic nutrients of an edible part of the 2,191 foods. Here, the average refuse rate was defined as “the portion of the food that is discarded in normal eating habits as a percentage of the whole food or purchased form” (MEXT, 2015). By combining this rate with the weight of food purchased by the households which was quantified in the original manuscript, we have estimated the weight of the edible part. The methodology is as follows.

First, we manually matched our most detailed food commodities based on the FIES (which is shown in Supplementary Table 4 in the SI) with the food items on STFC as detailed as possible. When the food commodity i on FIES can be attributed to food item(s) l , that is $i \in l$, the

inedible ratio on commodity i , ρ_i was calculated as:

$$\rho_i = \frac{\sum_l \rho_l}{n_i} \quad (S1)$$

where n_i denotes the number of attributable commodities. Hence, the above equation represents the arithmetic mean value of the refuse rates. We recognize that this method of taking averages may affect the determination of ρ_i , however, there is currently no information about consumption amounts of each food item according to STFC, to our best knowledge.

Therefore, we also prepared $\rho_{i,\max}$ and $\rho_{i,\min}$ by retrieving the highest and lowest inedible ratios among food items related to food commodity i .

$$\rho_{i,\max} = \max_{l \in I} \rho_l \quad (S2)$$

$$\rho_{i,\min} = \min_{l \in I} \rho_l \quad (S3)$$

For example, *sea bream (tai*, in Japanese) on FIES can be attributed to the four different species and two different types (natural or aquaculture) including fresh processing (i.e., sashimi) (therefore, six items on STFC are attributed to *sea bream* on FIES), ranging from 0 to 60% of the inedible ratios. Then, ρ_i , $\rho_{i,\max}$, and $\rho_{i,\min}$ represent 45.8, 60, and 0. Note that we have utilized ρ_i for presenting the results in the main text and both $\rho_{i,\max}$ and $\rho_{i,\min}$ for quantifying their uncertainty in terms of the inedible ratio.

Second, we have revised the average FW per expenditure by multiplying $(1 - \rho_i)$ by the right term of Equation 1 in the original manuscript; the new Equation 1 in the revised manuscript is presented as Equation S4.

$$w_{ik} = u_i q_i \sigma_i r_{ik} \quad (S4)$$

where $\sigma_i = (1 - \rho_i)$ represents the survival ratio. For the uncertainty analysis, we utilized w_{ik} when ρ_i were replaced with $\rho_{i,\max}$, and $\rho_{i,\min}$ in the above equation.

The calculation results using ρ_i were presented as the main result in the revised manuscript. For uncertainties associated with the inedible ratios, we set $\rho_{i,\max}$, and $\rho_{i,\min}$ as the lower and upper boundaries of the inedible ratio as well as the FW ratio. The sensitivity results were described in the SI as well as those of the FW ratio. Fig. R4 represents the ranges of predicted FW and FWGHG for 11 food categories based on their 95%CI as Fig. R4.

Fig. R4. Comparisons of household FW (a) and FWGHG (b) for the three different inedible ratios across food categories. “Mean” denotes the result of the mean inedible ratios. “Upper bound” and “Lower bound” denote the cases in which the inedible ratios were taken between the maximum and minimum values by food commodity on the STFC, respectively.

Importantly, these changes in food waste ratio based on FLSS did not seriously affect the discussion and conclusion already described in the original manuscript. Note that since the FW and FWGHG by commodity were not increased/decreased uniformly as the food waste ratio and inedible ratio due to the normalization of total FW, we do not show the calculation results as error bars. We have explained the above methods and all results of the predicted mean FW and FWGHG, these 95% CI in the Result section and the SI.

Finally, we recognize that, as pointed out by the other reviewer, it is important to incorporate the effects of changes in household income (or GDP per capita), food price, dietary trends by time, cohort, and so on for the projections of FW and FWGHG in the future. Although we could project the FW and FWGHG from 2020 to 2040 by applying the results of estimating food consumption expenditure that can be obtained from the report published by the Policy Research Institute of the Ministry of Agriculture, Forestry and Fisheries (PRIMAFF, 2019) to our FW and FWGHG intensities, we have determined to include the results in the SI (please see our reply to Comment #2 from Reviewer 2).

Based on these replies, we have added clarifications in the main text and the SI in the revised manuscript. Thank you very much for your constructive comments and kind guidance.

References:

- Consumer Affairs Agency, Government of Japan, 2023. Guidebook for Reduction of Food Loss and Waste, https://www.caa.go.jp/policies/policy/consumer_policy/information/food_loss/pamphlet/#guidebook (in Japanese) (Final accessed. Jan. 11, 2024)
- Crippa, M., Solazzo, E., Guizzardi, D., Monforti-Ferrario, F., Tubiello, F. N., Leip, A. J. N. F. (2021) Food systems are responsible for a third of global anthropogenic GHG emissions. *Nat. Food*, 2(3), 198-209.
- Kummu, M., De Moel, H., Porkka, M., Siebert, S., Varis, O., Ward, P. J. (2012) Lost food, wasted resources: Global food supply chain losses and their impacts on freshwater, cropland, and fertiliser use. *Sci. Total Environ.*, 438, 477-489.
- MEXT: Ministry of Education, Culture, Sports, Science and Technology, Japan (2015) STANDARD TABLES OF FOOD COMPOSITION IN JAPAN - 2015 - (Seventh Revised Version).
- Nansai, K., Fry, J., Malik, A., Takayanagi, W. & Kondo, N. (2020) Carbon footprint of Japanese health care services from 2011 to 2015. *Resour. Conserv. Recycl.* 152, 104525.
- PRIMAFF, 2019. Policy Research Institute, Ministry of Agriculture, Forestry and Fisheries, Estimated Future Food Consumption in Japan (2019), https://www.maff.go.jp/primaff/seika/attach/pdf/190830_1.pdf (in Japanese) (Final accessed. Jan. 11, 2024)
- Tantiwatthanaphanich, T., Shao, X., Huang, L., Yoshida, Y. & Long, Y. (2022) Evaluating carbon footprint embodied in Japanese food consumption based on global supply chain. *Struct. Chang. Econ. Dyn.* 63, 56–65.
- UN, 2021. United Nations Environment Programme. Food Waste Index Report 2021. Nairobi.

To Reviewer 2:

Comment

1. Reviewer thinks it is a novel research which estimates the food waste (FW) detailed by age brackets and conducts future estimation. However there still need improvement in the model.

Response:

Thank you for your kind support. We have addressed your invaluable comments and substantially improved our model as follows.

2. This model assumes that the consumption in each age bracket does not change through 2015 to 2040 except the effect of household size. This is a strong assumption because consumption will change by various factors such as cohort effect, trend of the times, price, income.

See The estimation of The Policy Research Institute of the Ministry of Agriculture, Forestry and Fisheries (PRIMAFF)

https://www.maff.go.jp/primaff/seika/attach/pdf/190830_1.pdf

(in Japanese)

Response:

We agree that your response and suggestions are crucial for our model for projecting the FW and FWGHG in the future. As you point out and via the suggested document, food consumption is affected by a change in various factors including household income, food price, dietary trends over time, cohort, etc. According to your advice, we looked into the suggested reference to consider the sensitivity of the result in this study. The report presents the estimation of changes in food consumption expenditures (monetary based) for three groups; fresh foods, processed foods, and restaurants from 2015 to 2040 (2015=100). In addition, it shows the compositions of food consumption expenditure for 11 categories every five years from 2015 to 2040, which are the same categories and analytical period adopted in this study. With these two pieces of information, we obtained increase/decrease ratios of the food consumption expenditures for the categories based on the 2015 values as shown in Figure R4.

Figure R4. Future trends in the food expenditure changes being estimated by PRIMAFF (2019) (2015=1)

Further, we attempted to project the FW and FWGHG from 2020 to 2040 by combining these ratios and the FW and FWGHG intensities by category obtained in this study if the food consumption amounts increase in line with the expenditures (therefore, the food prices are assumed to be constant from 2015) as is implied in the reference. Overall, the FW and FWGHG were expected to decrease markedly compared to the initial projections in this study. In 2040, FW and FWGHG were estimated at 2.43 Mt/year (-16.0% compared to 2015) and 5.79 Mt-CO₂eq/year (-9.9%), respectively. These reasons are explained mainly by expectations that dependencies on processed foods and restaurants would grow in contrast to fresh foods due to increases in single and elderly households who are more likely to prefer these foods (PRIMAFF, 2019). The detailed results are shown in Table R5 and Table R2. However, it is extremely difficult to reproduce the methodology to estimate future trends in food consumption expenditure since the detailed results of trends in food consumption expenditure and the material and methods are not sufficiently presented in the suggested reference.

Fig. R5. Future projections of household FW and FWGHG by using the food expenditure changes being estimated by PRIMAFF (2019) and these intensities calculated in this study.

Table R2. Future projections of household FW and FWGHG by using the food expenditure changes being estimated by PRIMAFF (2019) and these intensities calculated in this study.

FW [Mt]						
	2015	2020	2025	2030	2035	2040
Grains	0.17	0.17	0.16	0.15	0.14	0.13
Fishery and seafoods	0.15	0.13	0.11	0.10	0.08	0.07
Meats	0.08	0.08	0.08	0.07	0.07	0.07
Dairy products	0.16	0.16	0.16	0.15	0.16	0.15
Vegetables	1.23	1.21	1.15	1.08	1.01	0.96
Fruits	0.41	0.39	0.35	0.32	0.29	0.27
Oils and seasonings	0.19	0.19	0.19	0.19	0.20	0.20
Confectionery	0.09	0.09	0.09	0.09	0.09	0.09
Ready meals	0.21	0.22	0.23	0.24	0.25	0.26
Soft drink	0.14	0.15	0.16	0.17	0.18	0.19
Alcohols	0.06	0.06	0.05	0.05	0.05	0.04
Total	2.89	2.85	2.74	2.61	2.52	2.43

FWGHG [Mt-CO ₂ eq]						
	2015	2020	2025	2030	2035	2040
Grains	0.48	0.48	0.45	0.42	0.40	0.37
Fishery and seafoods	0.70	0.64	0.54	0.46	0.39	0.33
Meats	0.71	0.70	0.68	0.66	0.64	0.61
Dairy products	0.28	0.28	0.27	0.27	0.28	0.27
Vegetables	1.28	1.25	1.19	1.12	1.05	0.99
Fruits	0.42	0.40	0.36	0.33	0.30	0.28
Oils and seasonings	0.41	0.42	0.42	0.43	0.44	0.44
Confectionery	0.36	0.37	0.37	0.36	0.36	0.35
Ready meals	1.13	1.19	1.24	1.29	1.34	1.40
Soft drink	0.18	0.19	0.20	0.21	0.23	0.24
Alcohols	0.14	0.14	0.13	0.12	0.12	0.11
Total	6.10	6.04	5.86	5.67	5.52	5.38

On the other hand, it can be mentioned that we focused solely on the demographic effect on the food demand affecting the FW and FWGHG; namely, both the changes in the number of households and population when those are projected until 2040. This is due to the difficulty of

estimating changes in household income levels and food prices with rigid evidence compared to cohort changes. This approach is, therefore, to measure the potential impact of demographic change rather than to “precisely estimate” the future FW and FWGHG. Toward this end, our previous studies that were published in refereed journals (Shigetomi et al., *Environ. Sci. Technol.*, 2014; Shigetomi et al., *Ecol. Econ.*, 2015; Chapman and Shigetomi, *Sci. Total Environ.*, 2018; Shigetomi et al., *Environ. Res. Lett.*, 2020) utilized a similar approach.

Therefore, although we have decided not to replace the above estimation results based on the report of PRIMAFF with the main results of this study. Instead, we have highlighted this reference to show a future perspective of this study from Line 243. In addition, in line with your advice, and for thoroughness, we have detailed the results of the above approach in the Supporting Information (SI) as a complementary evidence base and sensitivity measure for this study. Thank you for your kind advice and understanding of our approach here.

3. The food waste ratio in FLSS indicate food loss (edible part) per food used (edible part). On the other hands , the weightin FIES and RPS is retail weight which may include inedible part. There seems no factor to fill the weight gaps.

Response:

We appreciate this comment on the methodology of quantifying the FW in this study. We confirmed that the FLSS indicates the intake and waste of *edible* food while our estimation of the amount of food consumption by household attribute focuses on a portion of a whole food (*edible + inedible*) by reviewing the document, as you indicated. Hence, to fill these weight gaps, we additionally referred to the database on the Standard Tables of Food Composition in Japan - 2015 - (Seventh Revised Version) published by the Ministry of Education, Culture, Sports, Science and Technology, Japan (MEXT). This database (hereafter, we call it STFC) has been published seven times since 1950 and the latest version is for 2020. We chose the tables for 2015 which is the closest to the analysis year in this study. It allowed us to obtain the average refuse rate as well as 52 basic nutrients of the edible part of the 2,191 foods considered. Here, the average inedible ratio was defined as “the portion of the food that is discarded in normal eating habits as a percentage of the whole food or purchased form” (MEXT, 2015). By combining this ratio with the weight of food purchased by the households which was quantified in the original manuscript, we have estimated the weight of the edible part. The methodology is as follows.

First, we manually matched our most detailed food commodities based on the FIES (which is

shown in Table S4 in the SI) with the food items on STFC as detailed as possible. When the food commodity i on FIES can be attributed to food item(s) l , that is $i \in l$, the inedible ratio on commodity i , ρ_i was calculated as:

$$\rho_i = \frac{\sum_l \rho_l}{n_i} \quad (\text{S1})$$

where n_i denotes the number of attributable commodities. Hence, the above equation represents the arithmetic mean value of the refuse rates. We recognize that this method of taking averages may affect the determination of ρ_i , however, there is currently no information about consumption amounts of each food item according to STFC, to our best knowledge. Therefore, we also prepared $\rho_{i,\max}$ and $\rho_{i,\min}$ by retrieving the highest and lowest inedible ratios among food items related to food commodity i regarding a sensitivity analysis indicated by the other reviewer (Comment #17).

$$\rho_{i,\max} = \max_{i \in l} \rho_l \quad (\text{S2})$$

$$\rho_{i,\min} = \min_{i \in l} \rho_l \quad (\text{S3})$$

For example, *sea bream (tai, in Japanese)* on FIES can be attributed to the four different species and two different types (natural or aquaculture) including fresh processing (i.e., sashimi) (therefore, six items on STFC are attributed to *sea bream* on FIES), ranging from 0 to 60% of the inedible ratios. Then, ρ_i , $\rho_{i,\max}$, and $\rho_{i,\min}$ represent 45.8, 60, and 0. Note that we have utilized ρ_i for presenting the results in the main text and both $\rho_{i,\max}$ and $\rho_{i,\min}$ for quantifying their uncertainty in terms of the inedible ratio.

Second, we have revised the average FW per expenditure by multiplying $(1 - \rho_i)$ by the right term of Equation 1 in the original manuscript; the new Equation 1 in the revised manuscript is presented as Equation S4.

$$w_{ik} = u_i q_i \sigma_i r_{ik} \quad (\text{S4})$$

where $\sigma_i = (1 - \rho_i)$ represents the survival ratio of food. The calculation results using ρ_i were presented as the main result in the revised manuscript.

For uncertainties associated with the inedible ratios, we set $\rho_{i,\max}$ and $\rho_{i,\min}$ as the lower and upper boundaries of the inedible ratio as well as the FW ratio. The sensitivity results were described in the Result section. Fig. R4 represents the ranges of predicted FW and FWGHG for 11 food categories based on their 95% CI as shown in Fig. R5.

Fig. R5. Comparisons of household FW (a) and FWGHG (b) for the three different inedible ratios across food categories.

As you can see Fig. R5, the uncertainties due to the inedible ratios look quite small, implying that the discussion and conclusion that have been already described in the previous manuscript are not seriously affected by these ratios. We have explained the above methods and all results of the predicted mean FW and FWGHG, these 95%CI outcomes are reflected in the Results section and the SI.

- Inventory data of IDEA for agricultural and fishery products are based on the weight at wholesale stage. However, weight and physical units used in this study is based on the retail stage. This may lead underestimation of FWGHG.

Response:

Thank you for reflecting this deep insight. As to this comment, it is necessary to additionally consider the life cycle GHG emissions related to agricultural and fishery products from wholesaler to retailer to avoid the underestimation indicated. However, it is extremely difficult to determine GHG emissions related to the transport of food from wholesale to retail by using IDEA. We, therefore, combined the information based on an environmental input-output analysis (EIOA) to fill the emission gap between wholesale and retail of the agricultural and fishery food wasted.

The Embodied Energy and Emission Intensity Data for Japan Using Input-Output Tables (3EID, Nansai et al., 2020) provides the direct GHG emission coefficients for commodities based on

Japan’s input-output table (JIOT). Combining JIOT and 3EID allows us to calculate the breakdown of emissions from the production to the retailer (i.e., emissions related to the margins). Since it is extremely difficult to transform the food weights at the retail stage that were estimated in this study to those at the wholesale stage, we calculated the emission ratio of the total emissions from the production to the retail to those from the production to the wholesale and multiplied it by the corresponding emission intensity (intensities) in IDEA. This procedure aims to complement the emission intensity on IDEA from wholesale to retail stages.

For this procedure, we have prepared a new equation for estimating the FWGHG for agricultural and fishery products ($i \in$ raw agricultural and fishery products) based on equation (4) in the original text by using the complemented emission ratio for the margin-related emissions from wholesale to retail stages, β_i , as follows.

$$\bar{F}_i^p = \bar{W}_i^p f_i(1 + \beta_i) \tag{4}$$

where f_i denotes the life cycle GHG emission intensity from raw material to wholesale on commodity i (i.e., IDEA). As a result, the total FWGHG increased by almost 125% in the revised manuscript compared to that in the previous manuscript (note that this increase includes not only the effect of considering the additional supply chain mentioned here but also the edible and non-edible parts of food as modified in your comment #3). In addition, the order of importance for the FWGHG were changed slightly (please see the second result section).

Additionally, for better understanding and clarity, we have added Fig. S1 which illustrates the system boundary of FWGHG in this study in the SI (here, Fig. R2).

Fig. R2. Food life cycle and the system boundary of the FWGHG in this study.

5. The low externality of food results in more food waste from elderly households. However, this can be said to result in less food waste from food services and other sources. Is there any implication to the national target for FW reduction?

Response:

We agree that this is also one of the important limitations that should be addressed in future studies. The NSFIE provides the consumption expenditures in the “general meals” sector (monetary unit) by age of the household head, suggesting that the older the age of the household head is, the lower consumption expenditures per capita on the restaurant is (see below Table R1). This implies that the potential food waste (and loss generated through the upstream supply chain) by the younger households is likely to be larger than that by the elderly households due to their higher demands if none of their dietary preferences at restaurants are considered. Besides, the amounts of food waste resulting from leftovers at restaurants (eating out), general banquet, and wedding parties were provided based on the survey sampling in 2015 (MAFF, 2015), accounting for 18.8, 298.8, and 336.7 grams per service (3.6%, 12.2%, and 14.2% of the total food served), respectively. In the same year, it was reported that the total amount of food waste from the food service industry was 1.33 Mt. However, there are difficulties in estimating the food waste by household brackets generated through food services and others for the following reasons:

There is no information about the sort of dishes and the amount of food served per service, its expenditure, and how many times the food service was used by households by the age of the household head in a year. These missing data do not allow us to estimate the food waste related to eating out by age bracket. We also think that assuming these to be the same among household brackets is too coarse to estimate the food waste regarding different dietary preferences.

Furthermore, it is impossible to identify the food consumption and the related waste in hotel services because the related consumption expenditure is aggregated into the expenditures in the “hotel” and “package tours” sectors on FIES and NSFIE. We cannot also consider the food waste that is generated in ceremonial scenes such as a wedding party because detailed household consumption expenditures on ceremonies by household do not exist (i.e., those are also aggregated into the expenditure on the “other miscellaneous” sector, see Table R2).

For these reasons, this study focuses solely on food waste from the home directly, although more food waste from elderly households would be generated compared to younger households due to the current focus on household food waste as per the reviewer's comment. We have detailed an analysis of the impact of FW on carbon emissions in the discussion in quantitative terms, particularly that food waste GHG emissions can be reduced via targeted actions in terms of selecting foods, particularly meats.

Table R2. Mean annual consumption expenditures per capita on food services among households of the age of householder in 2014 (MIC, 2021). Unit: JPY

Commodity	20s	30s	40s	50s	60s	70s
General meals	8,789	4,960	4,351	5,309	4,850	3,816
Hotel	893	537	428	807	850	824
Package tours	1,108	580	544	1,277	2,328	2,389
Other miscellaneous	3,385	3,237	3,057	4,880	5,651	6,045

References:

- Chapman, A. & Shigetomi, Y. Visualizing the shape of society: An analysis of public bads and burden allocation due to household consumption using an input-output approach. *Sci. Total Environ.* 639, 385–396 (2018).
- MAFF: Ministry of Agriculture, Report on Food Loss Statistics Survey 2015 (Food Service Survey) (2015) <https://www.e-stat.go.jp/stat-search/files?page=1&layout=datalist&toukei=00500231&tstat=000001015650&cycle=8&year=20151&month=0&tclass1=000001032628&tclass2=000001082415> (in Japanese)
- MEXT: Ministry of Education, Culture, Sports, Science and Technology, Japan (2015) STANDARD TABLES OF FOOD COMPOSITION IN JAPAN - 2015 - (Seventh Revised Version).
- Shigetomi, Y., Chapman, A., Nansai, K., Matsumoto, K. & Tohno, S. Quantifying lifestyle based social equity implications for national sustainable development policy. *Environ. Res. Lett.* 15, 084044 (2020).
- Shigetomi, Y., Nansai, K., Kagawa, S. & Tohno, S. Changes in the Carbon Footprint of Japanese Households in an Aging Society. *Environ. Sci. Technol.* 48, 6069–6080 (2014).
- Shigetomi, Y., Nansai, K., Kagawa, S. & Tohno, S. Trends in Japanese households' critical-metals material footprints. *Ecol. Econ.* 119, 118–126 (2015).

REVIEWER COMMENTS

Reviewer #1 (Remarks to the Author):

Thank you for the opportunity to review the revised version of the manuscript entitled "Curbing household food waste and associated climate change impacts in an aging society", which deals with a topical concern, namely the assessment of the greenhouse gas emissions associated to household food waste from raw materials to retail. Specifically, the research explored food waste in Japan, which is impacted by an aging population. The authors have substantially revised the manuscript according to the reviewer's suggestions and have enhanced its scientific soundness and its clarity.

In the abstract, the authors added a description of the methodology and the quantitative insights of the research, as requested, whereas in the "Introduction" the authors included statistics and facts related to food waste economic and environmental consequences, making it more consistent and useful for readers. Moreover, as an essential part, the authors have better justified the choice of Japan as a case-study area, highlighting that its food waste quantity is higher than other countries, therefore including a preliminary comparison of the research background with other international realities. Although "Results" were rather clear and consistent, the authors have enhanced their description by clarifying some important aspects related to the methodological approach. Furthermore, the authors have improved figures, which at current are self-explicative and clear to readers.

Reviewer #2 (Remarks to the Author):

In comment #4 of reviewer 2, authors improved to take GHG emissions in sectors between producers and consumers. I think it is a meaningful revision. However, I discussed at a different point in the last comment.

IDEA provides LCI data based on wholesale weight.

For example of beef, "boneless cut of beef " (in Japanese "bubun-niku") are provided.

Usually consumers do not purchase "boneless cut of beef " but "dressed meat"(in Japanese "sei-niku") . When retailers cut "boneless cut of beef "and make "dressed meat", weight decreases >20% (meat sinews and excess fat can be discarded).

We need to treat this weight change, because we should to make clear how we treated this allocation issue between edible part and inedible part.

To Reviewer 1:

Comment

1. Thank you for the opportunity to review the revised version of the manuscript entitled “Curbing household food waste and associated climate change impacts in an aging society”, which deals with a topical concern, namely the assessment of the greenhouse gas emissions associated to household food waste from raw materials to retail. Specifically, the research explored food waste in Japan, which is impacted by an aging population. The authors have substantially revised the manuscript according to the reviewer’s suggestions and have enhanced its scientific soundness and its clarity.

In the abstract, the authors added a description of the methodology and the quantitative insights of the research, as requested, whereas in the “Introduction” the authors included statistics and facts related to food waste economic and environmental consequences, making it more consistent and useful for readers. Moreover, as an essential part, the authors have better justified the choice of Japan as a case-study area, highlighting that its food waste quantity is higher than other countries, therefore including a preliminary comparison of the research background with other international realities. Although “Results” were rather clear and consistent, the authors have enhanced their description by clarifying some important aspects related to the methodological approach. Furthermore, the authors have improved figures, which at current are self-explicative and clear to readers.

Response:

We appreciate your kind words. Thank you so much again for your constructive comments throughout the revision process.

To Reviewer 2:

Comment

1. In comment #4 of reviewer 2, authors improved to take GHG emissions in sectors between producers and consumers. I think it is a meaningful revision. However, I discussed at a different point in the last comment.

IDEA provides LCI data based on wholesale weight. For example of beef, "boneless cut of beef" (in Japanese "bubun-niku") are provided. Usually consumers do not purchase "boneless cut of beef" but "dressed meat"(in Japanese "sei-niku") . When retailers cut "boneless cut of beef" and make "dressed meat", weight decreases >20% (meat sinews and excess fat can be discarded). We need to treat this weight change, because we should to make clear how we treated this allocation issue between edible part and inedible part.

Response:

Thank you for very specific comment on this issue which we had overlooked. As the previous revision clarified, our method quantifies the physical amount of food waste generated by households and these lifecycle GHG emission from the raw material to the retail stages. In this regard, we think the allocation issue between the edible and inedible part of food (meats) consumed by households is already considered. However, considering your critical advice, the current result of the FWGHG of meats does not consider the process of dressed meat (sei-niku), which may result in underestimation. However, the IDEA database does not provide the GHG emission intensities for dressed meats. Also, to our best knowledge, there was no direct data to fill the emission gap between the boneless cut of meat and dressed meat for beef, pork, and the other meats.

To fill these above emission gaps, we attempted to apply 3EID to the emission intensities for the boneless cuts of beef, pork, and other meats (that is named "meat (for consumption)" in the inventory), as demonstrated in the previous revision for some agricultural and seafood products. Specifically, we have determined the FWGHG for meat products ($i \in$ beef, pork, and other raw meats) based on equation (4) in the original text by using the complementary emission ratio for the margin-related emissions from wholesale to retail stages, β_i , as follows.

$$\bar{F}_i^b = \bar{W}_i^b f_i (1 + \beta_i) \quad (4)$$

where f_i denotes the life cycle GHG emission intensity from raw material to wholesale for commodity i (this equation and explanation is the same as the previous response #4 to

reviewer 2). This implies the dressed meat from the boneless cut of meat is produced at the retail stage. As a result, the GHG emission intensities for those meats increased by almost 13% and both the beef and pork increased their impact on the FWGHG. Although β_i associated with 3EID cannot distinguish the meat species (i.e., between beef and pork), we believe this revision may address your concern as much as possible at the current point in time.

Note that as we received this comment, we carefully re-reviewed the LCI and have corrected the emission intensity for chicken by replacing “Parts of meat other than beef and pork (ushi, buta igaino bubunniku)” with “Broiler processed products (incl. dismantled products)” because the latter intensity would be more suitable for chicken consumed by households and the former intensity is estimated by the arithmetic average of both “ushi bubunniku” and “buta bubunniku”. In addition, the latter intensity covers from the meat production to packaging, according to IDEA’s description. Thus, we did not apply the above eq. 4 to the FWGHG for chicken. This is the reason why the total FWGHG and that of the chicken shrink in this revision compared to the previous results. Through these revisions, the total FWGHG and the per-capita FWGHG by age group decreased (for example, from 6.16 Mt-CO₂eq to 6.00 Mt-CO₂eq by the total FWGHG and from 39.8 kg-CO₂eq/cap to 38.9 kg-CO₂eq/cap by their 20s’ and younger households).

On the whole, the main discussion and conclusions of this study remain stable. We have thoroughly revised the results and the related figures and tables in the main manuscript and the supplementary information. Thanks to your careful and constructive comments, we believe that the manuscript has been better clarified and improved. We really appreciate your kind assistance.

REVIEWER COMMENTS

Reviewer #2 (Remarks to the Author):

Thank you for your effort to improve the manuscript.

1. Regarding past review comment (1 of Reviewer 2), I pointed that f_i may change depends on the allocation methods between edible part and inedible part for some kinds of food. It related to GHG intensity of food production stage per weight of edible part. GHG emission after wholesale stage [β_i] are not concerns this discussion.

When weight ratio of inedible part at wholesale stage is a , corrected GHG intensity f'_i , the GHG intensity per weight of edible part at retail stage, should be below.

-Mass allocation:

$$f'_i = f_i$$

-All GHGs are allocated to edible part:

$$f'_i = f_i / (1 - a)$$

It shows that mass allocation is implicitly assumed in this manuscript.

However, mass allocation between edible part and inedible part seems not common assumption. Reviewer thinks authors need to clearly indicate allocation method in the manuscript.

2. In Suppl. Table 4, f_i can be easily calculated if Margin compensation [β_i] is zero. IDEA's terms and conditions prohibited to show emission intensity in the publication without permission. I afraid that this violates the term. Please check IDEA's terms and conditions and show the results carefully.

To Reviewer 2:

Comment

1. Regarding past review comment (1 of Reviewer 2), I pointed that f_i may change depends on the allocation methods between edible part and inedible part for some kinds of food. It related to GHG intensity of food production stage per weight of edible part. GHG emission after wholesale stage [β_i] are not concerns this discussion. When weight ratio of inedible part at wholesale stage is α , corrected GHG intensity f'_i , the GHG intensity per weight of edible part at retail stage, should be below.

-Mass allocation: $f'_i = f_i$

-All GHGs are allocated to edible part: $f'_i = f_i / (1 - \alpha)$

It shows that mass allocation is implicitly assumed in this manuscript. However, mass allocation between edible part and inedible part seems not common assumption.

Reviewer thinks authors need to clearly indicate allocation method in the manuscript.

Response:

Thank you for your detailed investigation of our previous reply to make the analysis of this study more rigorous. This additional suggestion helped us understand the true intent of the previous comment and review the allocation method of lifecycle GHG emissions for raw meat products; *beef*, *pork*, and *other raw meats* whose emission intensities as the “boneless cut of meat” at the wholesale stage on IDEA again. The MAFF ¹ reports a rough estimation of yield ratios for beef (Japanese beef: wagyu beef) and pork from these boneless cuts of meat to dressed meats, accounting for 90.9% (300 / 330 kg) and 83.3% (50 / 60 kg), respectively. However, the analyzed year for these ratios is not clarified (perhaps 2023 according to the webpage) and the past ones cannot be traced. Thus, we adopted these values to our analysis and recalculated the FWGHG under the assumption that the yield ratios in 2015 for *beef*, *pork*, and *other raw meats* are the same as the available publication ¹. The mass and life cycle GHG emissions at the wholesale stage are illustrated in Fig. 1.

Mass of a boneless cut of meat on the wholesale stage [M]

Fig. 1 A schematic figure of the calculation of life cycle GHG emissions for a boneless cut of meat on the wholesale stage. GHG , GHG^{edible} , and GHG^{inedible} represent the life cycle GHG emissions with respect to the mass of the meat. f represents the life cycle GHG emission intensity at the wholesale stage. M and α ($0 < \alpha \leq 1$) represent the mass of meat (including edible and inedible parts) and the yield ratio of the edible part at the wholesale stage, respectively.

As the reviewer suggests, when the life cycle GHG emissions of food at the retail stage should be attributed to its edible part, the adjusted emission intensity can be described as

$$f^{\text{adj}} = \frac{GHG}{M\alpha} = \frac{f}{\alpha}$$

(if inedible part is set as α instead of $1 - \alpha$, the equation will become the same as the one in your comment).

Now, $\alpha_{\text{beef}} = 0.909$ and $\alpha_{\text{pork}} = 0.833$. For *other raw meats*, the yield ratio $\alpha_{\text{other raw meat}}$ would need a consideration of other meats than *beef* and *pork* such as *lamb*. However, we could not find a good reference for this ratio and determined it the arithmetic sum of the yield ratios of beef and pork as the same method of calculating the emission intensity in IDEA. In addition, the yield ratios of these imported raw meats are assumed to be the same because it is not possible to distinguish between domestic and imported food as already mentioned in the limitations. Therefore, we have underpinned these assumptions as a new limitation and future direction in the methodology section in the supplementary information.

Conclusively, we have added the description of the above extension for some of the meats in the methodology section (L. 359-364 on p. 14), the limitation section (L. 446-447 on p. 17), and the supplementary information (L. 91-109 on p. 7). In addition, we have modified the relative results and discussion. The result of FWGHG was slightly increased from 6.00 to 6.06 Mt-CO₂eq/yr. The discussion of reduction in FWGHG by decreasing wasted meats has been

enhanced. We hope this revision is appropriate and in line with your advice.

1. Ministry of Agriculture Forestry and Fisheries of Japan (MAFF), The Situation Concerning Meat and Poultry Eggs: Distribution of beef and pork (in Japanese), <https://www.maff.go.jp/j/chikusan/shokuniku/lin/>, accessed on May 21, 2024

2. In Suppl.Table4, fi can be easily calculated if Margin compensation [β_i] is zero. IDEA's terms and conditions prohibited to show emission intensity in the publication without permission. I afraid that this violates the term. Please check IDEA's terms and conditions and show the results carefully.

Response:

Thank you for pointing this out. Yes, we do not have the permission and do not want to publish the emission intensity on IDEA without it. As per your suggestion, we have deleted all the sheets incl. Supplementary Table 4 where we recorded the most detailed FWGHG results that can give the emission intensities on IDEA by using the results of FW. The current supplementary file reports all the FW and some FWGHG in the aggregated categories, which do not violate the terms of IDEA.

REVIEWERS' COMMENTS

Reviewer #2 (Remarks to the Author):

I believe this manuscript is sufficiently high quality to publish. I have no further comments.